# Research

evolution, microbiology, genomics

cyanobacteria, molecular clock, horizontal gene transfer, Great Oxygenation Event, Archean, Cryogenian

**Author for correspondence:**
G. P. Fournier
e-mail: g4nier@mit.edu

# The Archean origin of oxygenic photosynthesis and extant cyanobacterial lineages

G. P. Fournier[1], K. R. Moore[1,2], L. T. Rangel[1], J. G. Payette[1], L. Momper[1,3] and T. Bosak[1]

[1]Department of Earth, Atmospheric and Planetary Sciences, Massachusetts Institute of Technology, Cambridge, MA, USA
[2]Planetary Science Section, NASA Jet Propulsion Laboratory, Pasadena, CA, USA
[3]Exponent, Inc., Pasadena, CA, USA

GPF, 0000-0003-1605-5455; LTR, 0000-0002-4481-8221; JGP, 0000-0001-6479-4557; LM, 0000-0002-1069-681X; TB, 0000-0001-5179-5323

The record of the coevolution of oxygenic phototrophs and the environment is preserved in three forms: genomes of modern organisms, diverse geochemical signals of surface oxidation and diagnostic Proterozoic microfossils. When calibrated by fossils, genomic data form the basis of molecular clock analyses. However, different interpretations of the geochemical record, fossil calibrations and evolutionary models produce a wide range of age estimates that are often conflicting. Here, we show that multiple interpretations of the cyanobacterial fossil record are consistent with an Archean origin of crown-group Cyanobacteria. We further show that incorporating relative dating information from horizontal gene transfers greatly improves the precision of these age estimates, by both providing a novel empirical criterion for selecting evolutionary models, and increasing the stringency of sampling of posterior age estimates. Independent of any geochemical evidence or hypotheses, these results support oxygenic photosynthesis evolving at least several hundred million years before the Great Oxygenation Event (GOE), a rapid diversification of major cyanobacterial lineages around the time of the GOE, and a post-Cryogenian origin of extant marine picocyanobacterial diversity.

## 1. Background

The evolution of oxygenic photosynthesis dramatically expanded the role of biological processes in Earth's geochemical cycles, altered the redox properties of Earth's surface and increased the fluxes within the surface carbon cycle [1,2]. The timing of the origin of this metabolism directly informs hypotheses about mechanisms that contributed to the rise of oxygen in the atmosphere and interpretations of biosignatures before the Great Oxygenation Event (GOE) (e.g. [3–7]). Temporal constraints on the subsequent phylogenetic and ecological diversification of oxygenic phototrophs also link geochemical and geologic events to the evolution of planktonic marine cyanobacteria (e.g. [8–11]) and photosynthetic eukaryotes (e.g. [12–16]).

Genomic data enable the reconstruction of phylogenies of gene and cellular lineages, including those of cyanobacteria and photosynthetic eukaryotes. When calibrated by fossils, phylogenies and sequence data also form the basis of molecular clock analyses, an important tool for investigating the early coevolution of Earth and life. The age of crown Cyanobacteria places a younger-bound constraint on the origin of oxygenic photosynthesis and is especially important for establishing narratives of planetary biogeochemical change. Due to the sensitivity of molecular clock analyses to sequence datasets, models and calibrations, and the complex

and often subjective criteria for selecting these, age estimates for crown Cyanobacteria have varied widely. Species tree molecular clock analyses have produced ages that predate the GOE by more than 1 Gyr (e.g. [17,18]), less than 0.5 Gyr [19–21], very close to the GOE [21,22] or even after the GOE [13,20,22]. These analyses vary in the sets of fossil calibrations used to establish minimum ages for cyanobacterial groups, including some very simple forms that may or may not represent extant members of crown-group Cyanobacteria [13,23–25]. Additional issues arise from the use of the GOE itself as a calibration point (see discussion in [19]) or calibrations that rely on interpretations of geological and geochemical proxies for the presence of oxygenic metabolisms, such as banded iron formations (e.g. [17,20]).

Reliable divergence time estimates for cyanobacteria require adequate sampling of modern organismal genomes, careful interpretations of the fossil record, multiple fossil calibrations and extensive tests of the impacts of evolutionary model parameters. Links between extant clades and the fossil record are particularly problematic: the taxonomic affiliations of all currently used cyanobacterial fossil calibrations have been questioned due to the absence of particularly diagnostic morphological characters [23,24] (electronic supplementary material, text S1). Fortunately, more robust and diagnostic Proterozoic cyanobacterial fossils do exist [23]: *Eoentophysalis* from the approximately 2 Ga Belcher Supergroup [26,27], *Obruchevella*, with the oldest preserved fossils reported in the silicified concretions of the 1.56 Ga Gaoyuzhuang Formation [28] and *Eohyella*, fossils of endolithic cyanobacteria from silicified ooids of the approximately 800 Ma Eleonore Bay Group [29]. These fossils contain diagnostic morphological characters that provide excellent opportunities for cyanobacterial molecular clock calibrations (see detailed description in electronic supplementary material, text S2). In this study, we sequenced the representatives of lineages morphologically similar to these fossils to bolster the placement of tightly coiled cyanobacterial fossils with *Spirulina*, and enable the use of *Eoentophysalis* and *Eohyella* as potential calibrations.

The cyanobacteria-derived plastids coevolve with the primary nucleocytoplasmic host lineages of photosynthetic eukaryotes, so the fossil records of these groups are commonly used to calibrate cyanobacterial evolution. Many of these fossils have more complex morphologies that can be assigned to modern chloroplast-containing eukaryotes, such as the Neoproterozoic fossils of red algae [30], green algae [31] and plants. These fossils can alleviate some of the issues described above, but studies based on them have to consider the impact of fast rates of sequence evolution within organelle lineages on molecular clock branch rate models [16].

Another source of dating information that has recently begun to be incorporated in molecular clock studies is horizontal gene transfers (HGTs). HGTs represent cross-cutting events between genome lineages and, therefore, can establish the relative ages of donor and recipient clades. This information has been used in several ways: propagating absolute date constraints along reticulated branches within gene trees [32,33]; establishing the temporal coexistence of lineages and consistency with fossil-calibrated divergence times [34]; co-estimating phylogenomic reconciliations and chronograms [35] or constraining the sampling of molecular clock posterior age estimates [19]. Much like the fossil record, these approaches all depend upon the quality of HGT inferences. These are frequently ambiguous due to lack of phylogenetic resolution, and gene tree–species tree reconciliation methods being sensitive to gene tree rootings, taxon sampling and model parameters [36].

Here, we present molecular clock analyses that test the impact of different sets of fossil calibrations on species tree divergence time estimates, including the inclusion of plastid lineages and different evolutionary rate models. We also demonstrate how the use of HGTs can improve model selection criteria and impose more stringent constraints on posterior subsampling for divergence time estimates. These analyses provide consistent support for an Archean origin of crown Cyanobacteria, and a long history of Archean biogenic oxygen production, as well as more precise age estimates for other key divergences in bacterial evolutionary history.

## 2. Results

To account for the numerous sources of uncertainty in cyanobacterial divergence time estimates, the following sets of conditions were tested: (i) inclusion of plastid lineages and calibrations across evolutionary models; (ii) impact of different cyanobacterial fossil calibrations; (iii) impact of different evolutionary models, including branch rate models and birth–death process priors. Additionally (iv) the impact of relative age constraints from 34 inter-phyla HGTs were tested, both as a means of evaluating evolutionary models, and as a set of constraints on the sampling of posterior age estimates.

### (a) Including plastid lineages and calibrations substantially impact cyanobacterial age estimates

Inclusion of red and green eukaryotic algal fossils as younger-bound calibrations had only a minor impact on crown cyanobacterial divergence times (table 1). However, these analyses also yielded very old ages for crown Archaeplastida, with mean ages ranging across models from 2.62 to 1.97 Ga. These age ranges were substantially older than the oldest ages estimated for crown Eukarya in fossil-calibrated eukaryal molecular clock studies that included similarly old algal fossil constraints (approx. 1.8 Ga) [40].

Even in the absence of eukaryotic fossil calibrations, the inclusion or omission of plastid sequences within the species tree substantially impacted divergence time estimates (electronic supplementary material, table S1) suggesting that the long branches within plastid lineages may result in an overestimation of divergence time estimates [16]. This effect was most pronounced for uncorrelated branch rate models, with mean ages of 2.58 Ga versus 3.07 Ga under a uniform (no_bd) prior, and 2.81 Ga versus 3.17 Ga under a birth–death (bd) prior (table 1). Autocorrelated models were more robust to the inclusion of plastid lineages. For example, under the CIR_nobd model, including plastid sequences increased mean age estimates by only 95 Ma (2.69 Ga versus 2.76 Ga), supporting the consistency and reliability of autocorrelated rate models when estimating divergence times within this dataset.

This comparison suggests that placing a secondary older-bound age constraint on Archaeplastida can enable cyanobacterial divergence time studies to make the best use of eukaryotic palaeontological data. In fact, in the absence of cyanobacterial fossil calibrations, constraining crown Archaeplastida to be younger than 1.8 Ga had a large effect on the observed crown age of Cyanobacteria across models.

**Table 1.** Impact of cyanobacterial fossil calibrations on divergence time estimates for crown Cyanobacteria. Fossil calibrations and key are described in electronic supplementary material, tables S2 and S3.

| crown Cyanobacteria age estimate (Ma) (mean (95% CI)) | | | | |
|---|---|---|---|---|
| evolutionary model | *Obruchevella Eohyella* (BA) | *Obruchevella Eohyella* plastid (BB) | *Obruchevella* endolith (BI) | *Obruchevella* endolith plastid[a] (BJ) | *Obruchevella Eohyella eoentophysalis* plastid[a] (BE) |
| UGAM bd | 3445 (3841–3066) | 2872 (3414–2567) | 3553 (4027–3173) | 2924 (3347–2628) | 3068 (3585–2787) |
| UGAM nobd | 3399 (3888–3109) | 2774 (3114–2524) | 3406 (3826–3093) | 2836 (3276–2574) | 2863 (3166–2656) |
| LN bd | 3188 (3528–2923) | 2875 (3253–2672) | 3226 (3627–2938) | 2978 (3437–2680) | 3054 (3416–2861) |
| LN nobd | 3142 (3457–2873) | 2749 (3062–2575) | 3220 (3556–2937) | 2896 (3208–2622) | 2947 (3351–2731) |
| CIR bd | 3266 (3589–3027) | 3136 (3365–2929) | 3278 (3681–3067) | 3163 (3463–2953) | 3208 (3477–3008) |
| CIR nobd | 3110 (3434–2908) | 2868 (3125–2680) | 3113 (3350–2892) | 2960 (3275–2780) | 2977 (3219–2789) |

[a]Plastid includes primary eukaryotic algal fossils, as well as secondary older-bound constraints on crown Archaeplastida. UGAM, uncorrelated gamma multipliers model [37]; LN, lognormal autocorrelated model [38]; CIR, Cox–Ingersoll–Ross process model [39].

Compared to analyses that used algal fossil calibrations only as younger-bounds (table 1), this additional constraint pushed mean crown Cyanobacteria ages 213–338 Ma younger under autocorrelated models (2.53–2.76 Ga), and up to 759 Ma younger under uncorrelated models (2.44–2.51 Ga).

## (b) Cyanobacterial fossil calibrations independently support similar Archean cyanobacterial crown ages

Cyanobacterial fossils *Obruchevella*, *Eohyella* and *Eoentophysalis* are well known but have had limited utility in molecular clock analyses because the proposed modern relatives of these organisms have been poorly represented in phylogenetic trees. To determine their potential for use as calibrations, we examined their likely placement in the tree, as informed by newly sequenced cyanobacterial genomes (as reported in [16] and the genomes sequenced for this study). Two tightly coiled cyanobacteria from marine microbial mats (BPC1_4624, BPC2_4625) were observed to group with the previously sequenced tightly coiled *Spirulina* sp. and not with the two loosely coiled sequenced *Arthrospira* representatives. The absence of the coiled morphology from other closely related cyanobacterial clades suggests the independent evolution of this character within these two groups. These molecular and morphological constraints motivated the use of the approximately 1.50 Ga tightly coiled *Obruchevella* [28] as the younger bound for the group containing the tightly coiled *Spirulina* sp. We did not explore the proposed application of this calibration to the *Arthrospira/Spirulina* ancestral node [23], because this would presume the coiled fossil morphology as ancestral to all cyanobacterial lineages within this morphologically diverse group, with multiple independent losses throughout.

The approximately 1.6 Ga endolithic fossil [41], and the 0.8 Ga *Eohyella* [29] have clear morphological similarities to extant endolithic cyanobacteria and their close relatives. Some previous studies have interpreted the approximately 1.6 Ga endolithic fossil as a member of Pleurocapsales (e.g. [42]). Six benthic, colonial coccoidal cyanobacteria from culture collections group with endolithic cyanobacteria within *Chroococcidiopsis/Chlorogloea* [16]. Therefore, we applied a younger-bound calibration of either 1.6 Ga (endolithic fossil) or 0.8 Ga (*Eohyella*) to this total group. In the absence of older-bound constraints on plastid ages, the age of crown Cyanobacteria did not substantially change in response to the inclusion of either fossil. In the presence of older-bound constraints on plastids, the inclusion of the older 1.6 Ga endolithic fossil calibration had a greater impact, increasing the observed mean age by approximately 100 Ma, from 2868 to 2960 Ma (figure 1 and table 1).

*Eoentophysalis* is the oldest diagnostic cyanobacterial fossil (approx. 2 Ga, with new-age estimates of 2.02–1.85 Ga [27,43]). We sequenced and analysed four metagenome-assembled genomes (MAGs) of pustular mat-forming, darkly pigmented coccoidal cyanobacteria that are considered to be morphological analogues of *Eoentophysalis* based on the specific division patterns and multilayered thick exopolysaccharide envelopes [44]. These MAGs grouped together in our phylogenetic analysis and formed a clade that was sister to *Spirulina* and its newly sequenced tightly coiled relatives (figure 2). However, the specific placement of the group represented by the *Eoentophysalis* fossil remains unclear. Therefore, we used this fossil as a younger-bound calibration on a group including several coccoidal cyanobacteria with multilayered envelopes that divide within the single envelope, a shared derived character found within both *Eoentophysalis* and this broader clade (figure 2, label 7). Including this *Eoentophysalis* fossil calibration recovered a mean age for crown Cyanobacteria of 2997 Ma. In the absence of plastid lineages and secondary older-bound calibrations, these ages shifted older by only approximately 100 Ma (table 1).

## (c) Evolutionary model choice is a major determinant of divergence time estimates

The selection of evolutionary model often substantially impacts results in molecular clock studies, and it is difficult to determine *a priori* which model best recovers the evolutionary rates across a tree. In the absence of criteria for preferring a particular model, a spread of age estimates across models should be considered, which substantially increases overall uncertainty [45]. The choice of evolutionary model is a complex and ongoing debate (see discussions in [39,46]) that

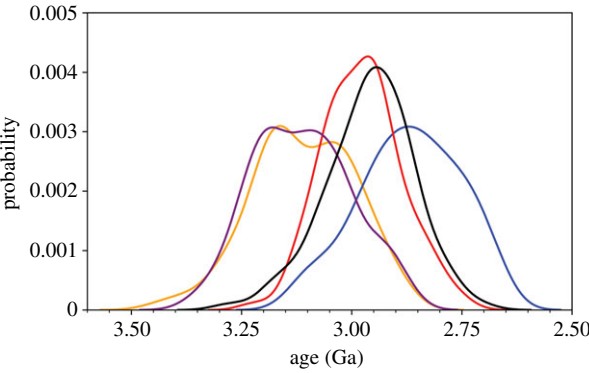

**Figure 1.** Posterior age distributions of crown Cyanobacteria using different sets of fossil calibrations. Model labels are as specified in table 1 and electronic supplementary material, table S3: BA = *Obruchevella* + *Eohyella* (orange); BB = *Obruchevella* + *Eohyella* + plastid (blue); BE = *Obruchevella* + *Eohyella* + plastid + *Eoentophysalis* (red); BI = *Obruchevella* + endolithic (purple); BJ = *Obruchevella* + endolithic + plastid (black). All ages shown are for the CIR_nobd evolutionary model.

includes considerations of uncorrelated (e.g. [37]) or autocorrelated (e.g. [38,39]) branch-specific rate models, as well as priors on the relative age distributions of divergences produced by speciation and extinction (birth–death) processes [46,47].

The evolutionary model parameters tested in this study produced highly variable divergence time estimates, both in the presence and absence of cyanobacterial fossil calibrations. In the absence of cyanobacterial or algal fossil calibrations, the 95% CI age range for crown Cyanobacteria across all models spanned approximately 1100 Ma (3654–2552 Ma) (table 1). The age range observed across all implemented fossil calibration schemas was approximately 1097 Ma across models, with mean values of approximately 1412 Ma for uncorrelated models, and approximately 939 Ma for autocorrelated models (table 1). Applying the most inclusive set of fossil calibrations (schema BE) resulted in crown Cyanobacteria ages across models spanning a similar age range of approximately 900 Ma (3585–2686 Ma) (table 1). Thus, for this study, the impact of fossil calibration choice is comparable to that of the choice of evolutionary model. The informativeness of the sequence data under the priors also varied across evolutionary models for key nodes within the chronograms (electronic supplementary material, figure S1).

## (d) Horizontal gene transfer-based constraints increase precision of divergence time estimates

To provide an empirical basis for the selection of evolutionary models, we used the relative ages of clades independently established by HGTs. We identified 34 strongly supported, manually curated 'index' HGTs between four phyla in the species tree used for this study. These HGTs represent 24 unique donor–recipient relative age constraints within the species tree (electronic supplementary material, table S4 and figure S2). Divergence times estimated under each evolutionary model were tested for consistency with these HGT constraints, in both the presence and absence of cyanobacterial fossil calibrations. These results identified the CIR process model combined with a uniform prior (CIR_nobd) as the most compatible in both the presence and absence of fossil calibrations; 75.79% and 65.09% of the posterior chronograms sampled under CIR_nobd fulfilled HGT

constraints for uncalibrated (BH) and calibrated (BE) age estimates, respectively. Other models showed consistently lower compatibilities, between 51.83% and 58.85% for uncalibrated estimates (BH), and between 44.45% and 56.86% for calibrated estimates (BE) (electronic supplementary material, figure S3). The CIR_nobd model also had the most HGTs consistent with at least 20% of the posterior space sampled (21 out of 24 for uncalibrated age estimates (BH), 19 out of 24 for calibrated age estimates (BE); electronic supplementary material, figure S4).

Only two HGTs showed zero or near-zero (less than 1.2%) compatibility across all models, both with crown Chlorobiaceae as the recipient clade. Following a previous study [19], we constrained the age of crown Chlorobiaceae with a younger-bound calibration based on 1.64 Ga isorenieratane carotenoid derivatives, as these lipids synthesized by a subset of extant members of this group [48] (electronic supplementary material, table S2). Because this inconsistency was closely associated with one of the few calibrated nodes within our analysis, we questioned the validity of this calibration. Removing the calibration from the divergence time estimation resulted in posterior ages with far greater compatibilities across HGTs where Chlorobiaceae is the recipient (electronic supplementary material, figure S5). This suggests that the calibration is phylogenetically misplaced: the ability to synthesize the observed 1.64 Ga biomarkers could have arisen earlier, either in stem Chlorobiaceae or in an entirely different clade of bacteria [49].

The preferred evolutionary model (CIR_nobd) dated crown Cyanobacteria to between 3275 and 2680 Ma, regardless of the specific set of fossil calibrations used (table 1, figure 3). The use of either the 1.6 Ga endolithic fossil (BJ schema) or 2.0 *Eoentophysalis* (BE schema) calibrations independently produced similar age range estimates for crown Cyanobacteria, with mean ages of 2960 Ma (95% CI 3275–2780 Ma) and 2977 Ma (95% CI 3219–2789 Ma), respectively (figure 2, table 1). When only eukaryotic primary and secondary fossil calibrations were used as calibrations, they recovered substantially younger, but still very likely pre-GOE dates for crown Cyanobacteria (95% CI 2733–2392 Ma). These age ranges were also consistent with those observed in previous similar molecular clock analyses that used a more controversial 1.6 Ga akinete fossil calibration [15,19]. The interpretation of these fossils as akinetes is questioned [23], but our HGT-constrained age estimates show the divergence of akinete-forming heterocystous cyanobacteria from basal heterocyst-forming types (represented in our tree by Rivularia) at 1177–1617 Ma, not inconsistent with this interpretation (figure 2).

Age estimates for total group Cyanobacteria were also similar under the CIR_nobd model, with the BE schema recovering a mean age of 3468 Ma with a 95% CI age range extending to 3716 Ma, and the BJ schema recovering a mean age of 3450 Ma, with a 95% CI age range extending to 3824 Ma. Finally, imposing index HGT constraints on the subsampling of chronograms used to calculate posterior ages increased the precision of age estimates across the tree, and shifted the mean ages for most nodes substantially younger (figure 2, table 2). For the BE schema, the resulting mean ages of crown Cyanobacteria under the CIR_nobd model shifted 87 Ma younger to 2900 Ma (95% CI: 2958–2774 Ma) (electronic supplementary material, figure S6), and total group Cyanobacteria shifted 81 Ma younger to 3387 Ma (95% CI 3496–3278 Ma).

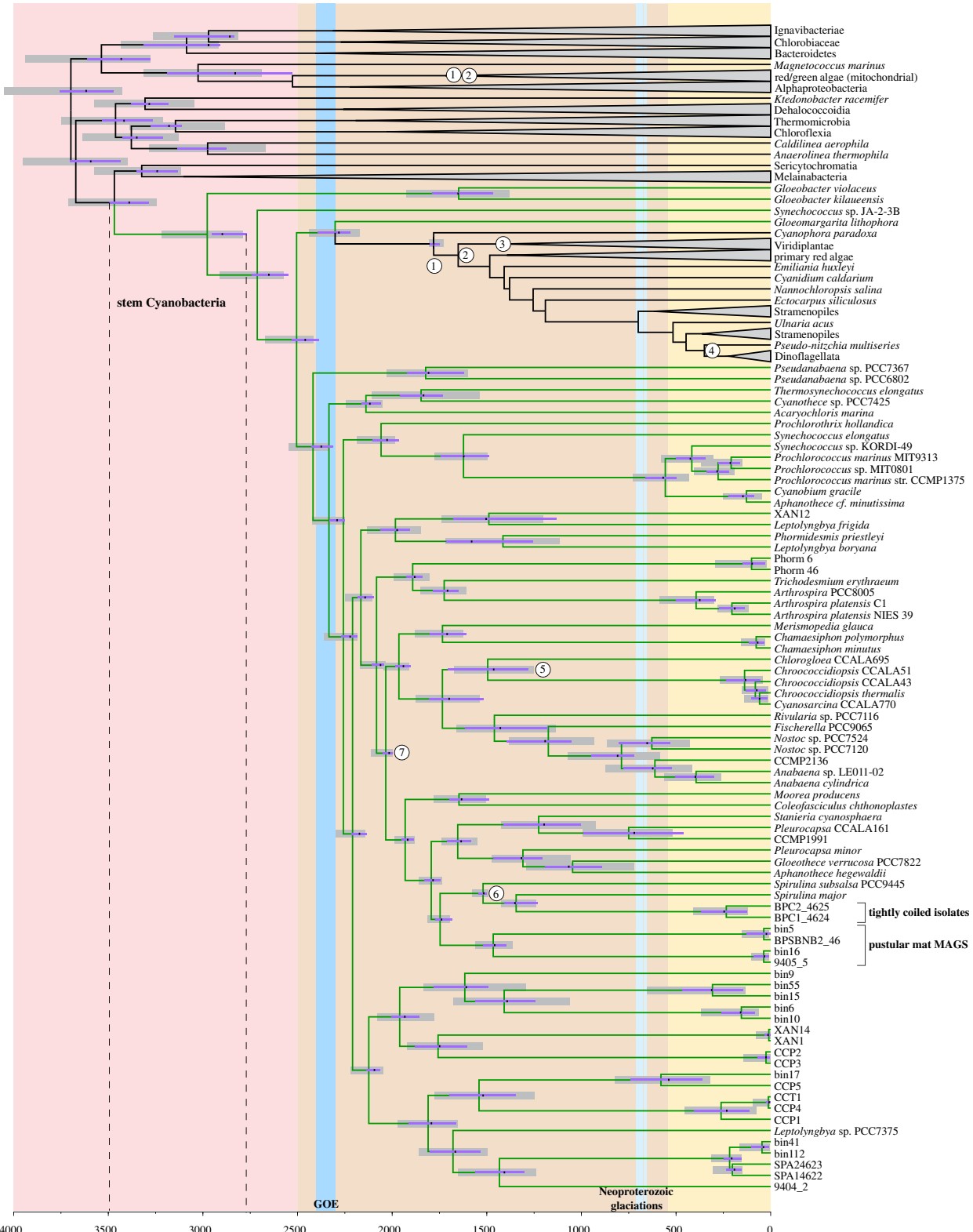

**Figure 2.** Chronogram of cyanobacteria. Grey bars show uncertainty (95% CI) under the CIR_nobd model using the BE calibration schema. HGT-constrained age estimates for each node are additionally included (black dashes indicate mean ages, purple bars indicate 95% CI uncertainty ranges). Major non-cyanobacterial clades are collapsed (grey triangles). Numbers indicate nodes of fossil calibrations (electronic supplementary material, table S2). Background colours represent Archean (red), Proterozoic (orange) and Phanerozoic (yellow) eons. The older and younger-bounds for the cyanobacterial stem lineage are indicated by dotted lines.

## (e) Post-Cryogenian origins of marine *Synechococcus* and *Prochlorococcus*

Marine *Synechococcus* and *Prochloroccus* species (the marine picocyanobacteria 'SynPro' group) are widespread, abundant and responsible for much of marine and planetary primary production [50,51]. Molecular clock analyses are especially important for understanding the evolutionary history of these groups because they do not produce a lipid biomarker record and have negligible preservation and fossilization potential [23]. Our analyses date the last common ancestor of extant marine SynPro to some time between the latest Cambrian (491 Ma) and the Middle Carboniferous (340 Ma) with a most likely age of 424 Ma (figure 2, table 2). This is

**Table 2.** Time estimates for a selection of key divergences under the CIR_nobd evolutionary model. Fossil calibrations are as described in table 1 and electronic supplementary material, table S3. 'Total' refers to total groups (inclusive of stem lineage).

| divergence time estimates: CIR nobd (Ga) (mean (95% CI)) | | | | | | | |
|---|---|---|---|---|---|---|---|
| | crown bacteria | total Cyanobacteria | crown Cyanobacteria | crown SynPro | total SynPro | crown Chloroflexi | crown Chloroflexia | total Chloroflexi |
| BE | 3.70 (4.05–3.42) | 3.47 (3.72–3.25) | 2.98 (3.22–2.79) | 0.42 (0.58–0.31) | 0.56 (0.73–0.43) | 3.46 (3.75–3.21) | 2.03 (2.61–1.49) | 3.15 (3.39–2.88) |
| BE + HGTs | 3.63 (3.76–3.48) | 3.39 (3.50–3.28) | 2.90 (2.96–2.77) | 0.42 (0.49–0.34) | 0.57 (0.67–0.49) | 3.42 (3.51–3.29) | 2.3 (2.47–2.10) | 3.18 (3.28–3.12) |

consistent with a previously published mean age estimate [52] but narrows the uncertainty and places this diversification squarely within the Palaeozoic. The divergence of total group marine SynPro from other groups (*Cyanobium* and related *Synechococcus*) most likely occurred during Late Ediacaran (571 Ma), potentially as early as the end of the Sturtian glacial period during the Cryogenian (666 Ma), or as recent as the Late Cambrian (494 Ma). This shrinks the previously estimated range, which extended from the Late Tonian (786 Ma) to the Early Carboniferous (355 Ma) [52]. In the absence of HGT constraints, our age estimate for this node also extends to the Late Tonian (726 Ma) (table 2).

## 3. Discussion

Analyses presented here consistently support the diversification of crown Cyanobacteria in the Archean, across several different evolutionary models and sets of fossil calibrations. The most constrained analyses under the preferred evolutionary model (CIR_nobd) date crown Cyanobacteria to the Late Mesoarchean or earliest Neoarchaean, with oxygenic photosynthesis predating the GOE by at least 360 Ma. If so, this lends independent support to some interpretations of the textural and geochemical Archean record as evidence for the presence of biogenic oxygenation (figure 4). These molecular clock analyses do not directly reveal how long cyanobacteria were oxygenic before their crown diversification because species trees do not currently resolve the evolutionary transitions between intermediate physiological states such as the integration of two photosystems with chlorophyll, or the incorporation of an oxygen-evolving complex. Any groups containing these intermediate sets of characters are either extinct or remain undiscovered. Under the additional and most parsimonious hypothesis that oxygenic photosynthesis originally arose in the cyanobacterial stem lineage following divergence from a non-photosynthetic sister group (specific scenarios are discussed in [53–55]), the age of total group Cyanobacteria serves as an older-bound on the origin of oxygenic photosynthesis. Alternatively, studies of individual photosystem genes have led to other, more complex scenarios, that hypothesize oxygenic phototrophy as more ancestral within bacterial diversity [21]. Under this proposed schema (figure 4), these dates suggest that a diverse and ecologically robust microbial ecosystem existed on Earth long before the evolution of oxygenic phototrophy. Inclusion of additional deeply branching phyla and index HGT constraints, or new evolutionary rate models could potentially further increase the precision and robustness of these age estimates. Finally, since the primary fossil calibrations used here are only younger-bounds, future discoveries of even older diagnostic fossils may provide additional evidence for crown Cyanobacteria and oxygenic photosynthesis during the Archean.

Our study suggests that the deepest branching extant cyanobacterial lineages (Gloeobacterales and the A/B' lineages of thermophilic Synechococcus) likely diverged by 2.7 Ga. Based on the estimated ages of the deepest branching extant cyanobacterial lineages (figure 4), the 2.7 Ga fossils and stromatolites attributed to benthic filamentous cyanobacteria [4,56] likely represent extinct or undiscovered groups that are distinct from later-diverging known groups of filamentous cyanobacteria. Our age estimates for crown Cyanobacteria are also in line with the proposed geochemical evidence for

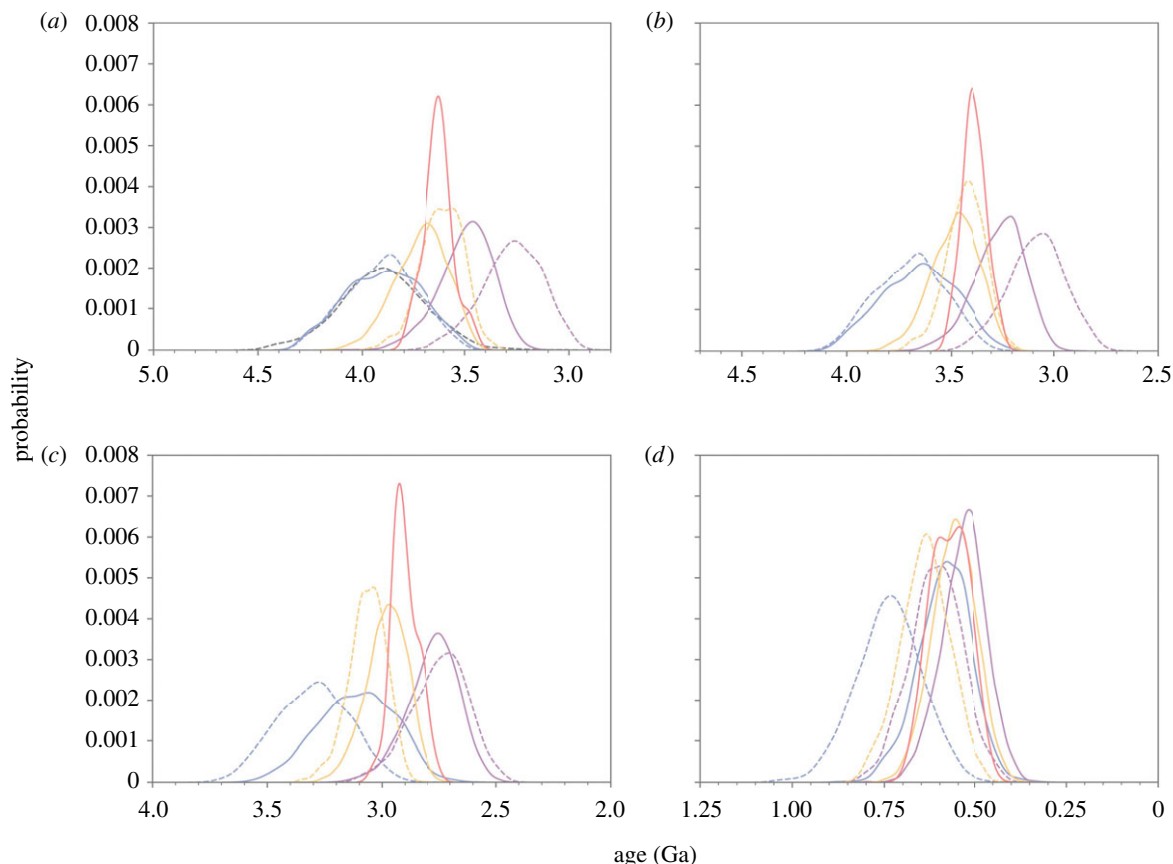

**Figure 3.** Divergence time estimates for major nodes in cyanobacterial evolution. (*a*) Crown bacteria (root); (*b*) total group Cyanobacteria; (*c*) crown Cyanobacteria; (*d*) total group marine SynPro. Prior (dashed) and posterior (solid) age distributions are shown. The root prior (black) is shown for crown bacteria (*a*). Age distributions are shown for the following analyses: uncalibrated (blue); calibration schemas excluding cyanobacterial/plastid calibrations (BH, purple); and including the cyanobacterial calibrations (BE, orange). Additionally, HGT-constrained age distributions are shown (red).

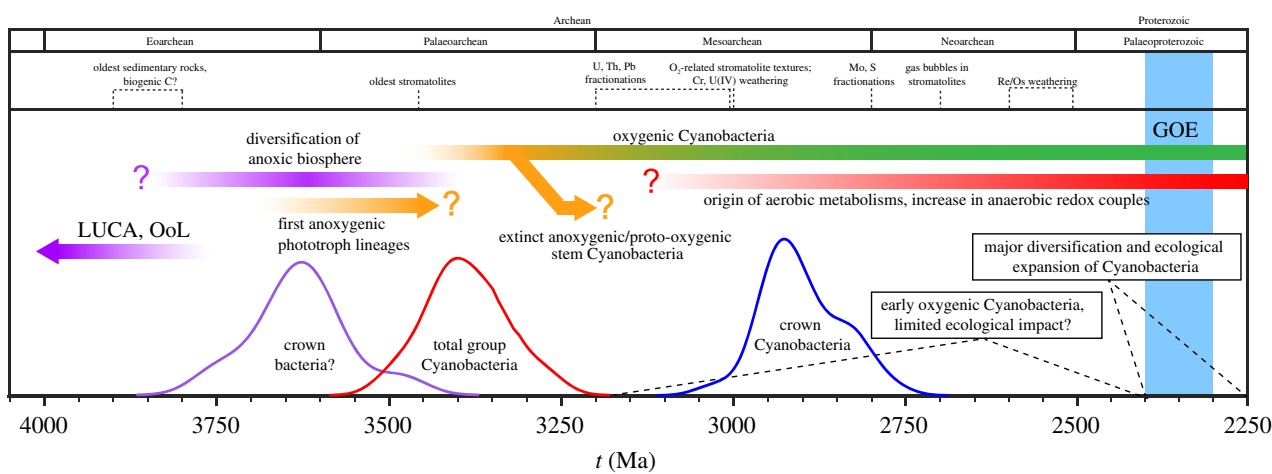

**Figure 4.** Proposed narrative for the cyanobacterial context of Earth's oxidation. Relative probability density distributions are shown for crown bacteria (purple), total group Cyanobacteria (red) and crown group Cyanobacteria (blue). This hypothesized history of cyanobacteria shows oxygenic forms arising from within a diversity of anoxygenic stem groups in the Archean. A timeline of geological evidence shows that these dates are consistent with Mesoarchaean and Neoarchaean geochemical and morphological evidence of oxygenic photosynthesis (detailed in electronic supplementary material, text S5).

localized and/or transient environmental oxygenation in the Late Archean (e.g. [2]). The age estimate for total group Cyanobacteria extending to approximately 3.50 Ga also allows for interpretations of textures in approximately 3 Ga Archean stromatolites from the Chobeni Formation [57] and some Archean trace metal oxidations [58,59] as oxygen-related, although less confidently so [7]. These age estimates are sufficiently restrictive to argue against a default cyanobacterial interpretation of all geochemical and morphological biosignatures older

than approximately 3 Ga. Instead, we interpret these results as being most consistent with primary producers in shallow water environments before approximately 3.4 Ga being early anoxygenic phototrophs of a likely extinct bacterial lineage, rather than oxygenic cyanobacteria, anoxygenic stem cyanobacteria or ancestors of the current anoxygenic phototrophic lineages (figure 4).

These age estimates suggest that early crown-group Cyanobacteria had little global atmospheric impact. By

contrast, the most abundant and diverse extant cyanobacterial clades do appear to radiate around the time of the GOE (figure 2). This supports a scenario in which ecological expansion and environmental diversification of cyanobacteria increased global carbon fixation and oxygen production at this time, in concert with additional geochemical, tectonic, atmospheric and microbiological processes (e.g. [60]).

These results also inform more recent periods of Earth history. Our results place the divergence of marine SynPro during or shortly after the Cryogenian glaciations. These periods of intense glaciation may have wiped out more ancient groups of marine cyanobacteria, with marine SynPro simply being the most recent ecological successors in this niche, along with eukaryotic algae [61,62]. Such patterns of extinction and succession within cyanobacteria would complicate attempts at ancestral environmental reconstruction based on the environmental distributions of extant species (e.g. [15]).

In summary, results obtained without using any calibrations that arise from geochemical hypotheses can be used as independent tests of the proposed geochemical evidence for the history of oxygenic photosynthesis. Our conclusions promote a view of cyanobacterial evolution that takes into account the extinction and succession of microbial groups, and an awareness of ecological prominence, rather than evolutionary origin, as a driving force behind planetary biogeochemical change.

## 4. Methods

### (a) Collection, enrichment, genome sequencing and assembly of the metagenome-assembled genomes of modern benthic cyanobacteria

Pustular and tufted microbial mats made by coccoidal cyanobacteria and other organisms were collected from Hamelin Pool near Carbla Station in Western Australia. These samples were collected under a Regulation 17 license, number 08-000373-1. Sample collection, enrichment of coccoidal cyanobacteria that form pustular mats and growth conditions are described in [44]. Detailed culturing, DNA extraction, sequencing, metagenome assembly and completeness/contamination testing methods are provided in electronic supplementary material, text S3. Following successful sequencing and reconstruction of cyanobacterial MAGs (greater than 97% complete and less than 3% contamination), ribosomal protein sequences used in [16] were identified and extracted from the resulting genomic assemblies using BLASTX.

### (b) Species tree reconstruction

The species tree used in this study was based upon the tree published in [16] using identical taxon sampling, with the addition of the newly sequenced strains described above (electronic supplementary material, table S5). Homologous ribosomal protein sequences identified from these newly sequenced genomes were added to each of the 30 ribosomal protein alignments, and re-aligned with MUSCLE v. 3.8.31 [63] and concatenated using FASconCAT v. 1.0 [64]. Best-fitting substitution model analyses were carried out using ProtTest v. 3.4.2 [65,66]. Maximum-likelihood trees both including and excluding plastid sequences were made with RAxML v. 8.1.9 [67] using the best fitting model parameters (four gamma-distributed site rate categories with estimated shape parameter ($\alpha = 0.827794$) and an LG substitution model).

Concatenated alignments and species trees are provided in electronic supplementary material, dataset S4 [68].

### (c) Detection of index horizontal gene transfers

Inter-phylum index HGT candidates were identified using a distance-based metric derived from the eggNOG 5.0 database [69], with subsequent phylogenetic analysis of candidate gene families. A detailed description of this approach is provided in (electronic supplementary material, text S4). Thirty-four index HGT events were identified that clearly constrained relative ages of donor and recipient clades within different phyla on the species tree, representing 24 unique pairs of donor and recipient clades (electronic supplementary material, table S4). Figures of trees containing labelled donor–recipient index HGT pairs are provided in electronic supplementary material, dataset S1 [68], and alignments and gene trees for each inferred index HGT are provided in electronic supplementary material, datasets S2 and S3 [68].

### (d) Molecular clocks and fossil calibrations

All primary fossil calibrations were applied as hard younger bound minimum age constraints (electronic supplementary material, table S2). Secondary calibration for the last common ancestor of plastids was applied as a range between 1050 Ma for the younger bound (the age of *Bangiomorpha* [30]) and 1800 Ma for the older bound, as estimated in [62]. Additional primary calibrations for plastid groups were also used, consisting of green algal and dinoflagellate fossils (electronic supplementary material, table S2). Other calibrations applied to outgroups are the same as in [19]. The root prior on the bacterial ancestor was normally distributed with a mean age of 3900 Ma and standard deviation of 200 Ma, in order to approximate a plausible range of hypotheses for the age of the bacterial root between 4300 and 3500 Ma [19]. Hard bounds were not included on the root prior with the goal of identifying model parameters and calibrations that push the root age estimate beyond plausible constraint (i.e. the age of the Earth) as a further means of model evaluation. Hard bounds at 4.4 Ga and 3.5 Ga were included on the root prior when estimating prior and posterior age estimates in the absence of any other calibration.

Divergence time estimates were calculated with Phylobayes 4.1c, using C20 substitution models, birth–death and uniform tree priors, and uncorrelated gamma distributed, autocorrelated lognormal and autocorrelated CIR process rate models for both prior and posterior estimates [70]. Two chains were run for each analysis using the default settings for the automatic stopping rule. For each analysis, at least the first 20% of chains were excluded as burn-in before the sampling of prior and posterior trees and calculation of age distributions in the chronogram (1 k trees for uncorrelated models, and 10 k trees for autocorrelated models). Chronograms and dates output files are available for each evolutionary model and set of calibrations in electronic supplementary material, dataset S5 [68].

### (e) Application of horizontal gene transfer-based age constraints

HGT-constrained age estimates were generated for the BE and BH calibration schemas. As provided by the 'datedist' output file from PhyloBayes, chronograms sampled after burn-in each represent a likely hypothesis under the given evolutionary model. For each HGT constraint, 'compatible' chronograms show the donor being older than the recipient; the percentage of compatible chronograms was then calculated for each HGT constraint (electronic supplementary material, figures S3 and S4). To estimate a posterior age distribution for chronograms

compatible with as many HGT constraints as possible, sets of trees passing $n$ HGT constraints were compiled, from $n = 1$ to $n = 24$ (electronic supplementary material, figure S6). Posterior age estimates were then calculated from the $n = 21$ set of trees (figures 2 and 3). This used the largest number of HGT constraints that recovered a set of compatible chronograms large enough ($n = 28$) to estimate age distributions. This method is robust against the impact of a single HGT constraint or biases arising from the stepwise addition of constraints given that each set of $n$ constraints can include any combination of HGTs.

Data accessibility. Supporting data files are available from the Dryad Digital Repository: https://doi.org/10.5061/dryad.x69p8czhm [71].

The data are provided in electronic supplementary material [72].

Authors' contributions. G.P.F.: conceptualization, data curation, formal analysis, funding acquisition, investigation, methodology, project administration, resources, software, supervision, validation, visualization, writing—original draft, writing—review and editing; K.R.M.: conceptualization, data curation, formal analysis, investigation, resources, writing—original draft; L.T.R.: data curation, formal analysis, methodology, software, supervision, validation; J.P.: data curation, formal analysis, investigation, methodology, software, validation, visualization; L.M.: data curation, formal analysis, investigation, methodology, resources; T.B.: funding acquisition, investigation, methodology, project administration, resources, supervision, writing—original draft, writing—review and editing.

All authors gave final approval for publication and agreed to be held accountable for the work performed therein.

Competing interests. We declare we have no competing interests.

Funding. This work was supported by the Simons Foundation Collaboration on the Origin of Life award grant no. 339603 to G.P.F. and award grant no. 327126 to T.B. and NSF Integrated Earth Systems award EAR grant no. 1615426 to G.P.F.

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
