## [Peer Review File · Proceedings of the Royal Society B: Biological Sciences]

Review History

RSPB-2021-0675.R0 (Original submission)

Review form: Reviewer 1 (Tanai Cardona)

Recommendation

Major revision is needed (please make suggestions in comments)

Scientific importance: Is the manuscript an original and important contribution to its field?

Excellent

General interest: Is the paper of sufficient general interest?

Excellent

Quality of the paper: Is the overall quality of the paper suitable?

Good

Is the length of the paper justified?

Yes

Should the paper be seen by a specialist statistical reviewer?

No

Do you have any concerns about statistical analyses in this paper? If so, please specify them explicitly in your report.

No

It is a condition of publication that authors make their supporting data, code and materials available - either as supplementary material or hosted in an external repository. Please rate, if applicable, the supporting data on the following criteria.

Is it accessible?

Yes

Is it clear?

Yes

Is it adequate?

Yes

Do you have any ethical concerns with this paper?

No

Comments to the Author

Fournier et al. present an excellent, comprehensive, and exciting molecular clock analysis tackling the issue of when the last common ancestor of cyanobacteria occurred. The authors enable the use of novel fossil calibrations by sequencing new genomes strategically located in a phylogeny of bacteria. And in addition, they implement an approach to improve the “precision and accuracy” of divergence time estimation by further constraining the clocks with a substantial number of horizontal gene transfer (HGT) events.

The authors conclude that the last common ancestor of cyanobacteria occurred well before the Great Oxidation Event, with the divergence of photosynthetic and non-photosynthetic cyanobacteria occurring between 3.25 and 3.50 Ga in their best preferred model. These observations were well supported by the presented data.

I will point out a few things that should be presented or discussed more adequately. I will also challenge a number of implicit assumptions that are not necessarily supported by the presented data or the current state of knowledge, but that in my opinion, do not take away from the quality of the work and results or the main conclusion regarding timing the last common ancestor of cyanobacteria. I wish my feedback to be taken as an ongoing and engaged discussion and a conversation, as I believe all my points can be reasonably addressed in a round of revisions.

I appreciate the authors making all alignments, trees, dates and rates files available for peer-review. That was a real treat and I enjoyed myself exploring these!

Overall, I think it is a fantastic effort. Thank you.

Key technical concepts

Line 24 and line 51, but see also further below. I think it is important to decouple the concept of “oxygenic photosynthesis” and the concept of “cyanobacteria”. They are not the same thing and their evolution is not coincidental in time. As our understanding of the evolution of photosynthesis advances, it is important that more precision is used to describe what is a nuanced evolutionary process.

There are four key elements here: 1) The last common ancestor of cyanobacteria giving rise to crown-group cyanobacteria, 2) the last common ancestor of melainobacteria (Vampirovibrionia) and cyanobacteria. These two are well defined points in the species tree of bacteria. 3) Stem-

group cyanobacteria, which represent ancestors located along the branch connecting the last common ancestors 1) and 2). And 4) the origin of oxygenic photosynthesis, which cannot be precisely located in a species tree.

You mostly use “stem cyanobacteria” to refer to point 2) when you report specific dates, but you also use it as in meaning 3). For example, as in line 318: “Molecular clock analysis does not directly reveal how long stem cyanobacteria were oxygenic before their crown diversification.” Importantly, point 1) and 2) represent a precise point in time, while meaning 3) represents a period of time, a series of undefined ancestors. Thus, as used in 318, it becomes a bit confusing.

Therefore, when you say in the abstract, “these results support oxygenic photosynthesis evolving several hundred million years before the Great Oxygenation Event (GOE)”, what do you actually mean? Do you mean that the results support a last common ancestor of cyanobacteria evolving several hundred million years before the GOE? Or do you mean that your data supports oxygenic photosynthesis evolving at least several hundred million years before the GOE? Given that the mean age for “stem cyanobacteria” under the preferred scenario (CIR nobd, BE+HGT) is a billion years before the GOE... The way it is expressed in line 24 in the abstract, it feels like you are using “oxygenic photosynthesis” and “crown cyanobacteria” as synonymous.

However, in line 51, it feels like the authors are using “oxygenic photosynthesis” and “stem cyanobacteria” as synonymous. Also note that in reference [17] and [18] it is actually “crown cyanobacteria” that are reported to be over a billion years after the GOE! Not just “stem cyanobacteria”.

I would like to encourage you to be a bit more mindful in the way these different concepts are used through the text. I also want to strongly recommend that you explicitly define in the introduction the terminology that you are going to use through the text; do not just assume the reader knows the way you personally like to refer to things. Tweak and refine as to be precise with what you mean, and double check for consistency.

Assumptions

This leads me to an important aspect of the way the data is interpreted. There is an implicit assumption that oxygenic photosynthesis originated after the divergence of melainabacteria. This is by no means a fact. That the last common ancestor of cyanobacteria and melainabacteria was non-photosynthetic is not a fact, it is only a hypothesis. When we bring the evolution of the photosystems into the picture, then it is also quite possible that oxygenic photosynthesis predates what you call “stem cyanobacteria” [1]. A recent analysis of the origin of oxygen-using enzymes is also consistent with photosystem evolution [2], echoing results from nearly a decade ago [3].

I do not think it is necessary for you to discuss these aspects in your paper in any detail. However, I do think you should acknowledge and state your assumption in a direct, clear and explicit way in the introduction, discussion, or where you find it more suitable.

Other comments

Line 53, “Much of this incongruity”. How much? I personally think the incongruity is mostly due to what you said in lines 48 to 50: a combination of factors. Perhaps rephrase?

Line 72, do you mean SI text 2? It is not clear. It would be a nice gesture for the readers if you could perhaps link the narrative of the SI text in a more fluid way to the main text. There is an abrupt change of pace when you shift from main to SI text, in such a way that I am confused as to whether you sent me to the right place: in particular, SI text 1, 2, and 5. Just add an introductory or linking opening statement. Thank you.

It would be nice too if you give the SI texts titles that reflect the content of the text.

End of introduction and start of results section. As a matter of convention, it is helpful to have a little paragraph at the end of the introduction that links to the results section, telling in a few brief

statements what you did, your rationale, and/or what you found. This is particularly important given that the introduction is not followed by the methodology.

The results section also opens in a way that it is not clear what is the question that you are going to address, what is the system that you are working with, or the hypothesis that you will be testing. As it is now, it is just really hard to penetrate and you will have the reader jumping all over the place trying to deduce or guess what you mean or did.

Line 116, the first data you refer to in the text is SI Table 1. When I went to find that table, I found DD, BH, +older bounds, bd, no bd. At this point none of these have been introduced to the reader. I am left then trying to find what these are: it took me a while to find out that the key with the experiments is found in SI Table 3, which also use some code of white, grey, filled boxes with numbers or an x, with no explanation.

I encourage the authors to reconsider how the data is presented, so that it makes for a smoother reading. To begin with, it is better if the first data that you choose to show is in the main text and not in the supplements. Perhaps you can structure things such that you first direct the reader to Table 1.

Your first paragraph of proper results, starting in line 111, is also a bit messy and hard to disentangle. Consider presenting the data first, then presenting the interpretation. For example, “under uncorrelated branch rate models the ages are X and Y. In contrast, under autocorrelated models the ages are A and B. We think these differences are caused by factors M and N, as proposed in reference [16].”

Line 120-123. It is stated that uncorrelated models independently assign higher penalties across branches inferred to be evolving at faster rates. What do you mean with “independently”? Is this something that you know from how the algorithms were developed or is this your rationalization of the observed effects? Can you provide a reference where these technical aspects of rate models are presented?

One major difference between uncorrelated and autocorrelated is that the former assumes an average rate for each internal node, while in the latter, rates at the internal nodes can vary to any magnitude. Is that linked to the penalties you talk about? Perhaps you want to keep it simple and elaborate on this on a supplementary text for the more specialist audience. Consider that the broader readership of Proceedings B interested in the evolution of cyanobacteria and photosynthesis may not be aware of what autocorrelated, uncorrelated, and birth-death means.

You use the expression, “far more robust”, what is your measurement for robustness? How much robust exactly is “far more”?

You conclude in the same paragraph stating that autocorrelated models are more consistent and reliable than uncorrelated. When looking at the patterns of rates of evolution from the datasets provided, I noted a substantial decrease in the rates of evolution with decreasing geological time, and a zone of large scatter after the GOE. I was not able to associate the rates to taxa, given that a file with labelled tree nodes was not provided (can you make these also available?), but I wonder if you could describe and discuss the patterns of rate change that you observe in your preferred scenarios as a supplementary text. I would appreciate that.

Section “Impact of eukaryotic calibrations”. I think the data and interpretation presented in this section can be better understood in terms of effects on the rates of evolution, rather than effects on divergence times. It is known that eukaryotic sequences evolve relatively faster than those in cyanobacteria, as clearly seen in the trees you provided in Dataset S1: the “long branches within plastid lineages”. When you add or remove taxa and calibrations as you describe, do these translate into increases or decreases on the rates of evolution?

For example, lines 141 to 150. In the absence of cyanobacterial calibrations and constraining photosynthetic eukaryotes to younger than 1.8 Ga, it is expected that this will translate into an acceleration of the rates of evolution for cyanobacteria, because the calculation of the rates will be dependent on the “younger than 1.8 Ga calibration”, which is allocated to the faster evolving eukaryotic sequences embedded within cyanobacteria. Therefore, it is expected that the measured ages for crown Cyanobacteria will appear younger given the calculated faster rates of amino acid substitutions.

Consider rephrasing for clarity.

Line 137. “Far more robust”. You make it sound like timing the evolution of eukaryotes with molecular clock is a done deal, or that the fossil record of eukaryotes is well understood. I am sure you recognize that molecular clocks have also generated massive uncertainties of nearly a billion years for crown eukaryotes. And that there are a bunch of eukaryote-like fossils and trace fossils that are older than 2.0 Ga. Perhaps you want to be a bit more cautious about how you make an interpretation of the utility of including plastid lineages in this type of analysis.

In any case, it is not clear what the take-home message of this section is regarding timing the evolution of cyanobacteria. Perhaps you wish to restate or rephrase for additional impact.

Line 152, Section (2). I think enabling these new calibrations is amazing. I wonder if you have images of the microbial mats where the MAGs came from, or microscopy images of the morphological analogues of *Eoentophysalis*, or the tightly coiled cyanobacteria strains? It would be great if you could share some of these in the supplementary materials.

Section starting in line 224. I want to say that I am indeed very positively impressed by the HGT-enhanced clock results and the effect the approach has on narrowing the confidence intervals.

I think it would be very useful to the reader if you can provide in a separate supplementary figure, the HGT transfer events that were studied, overlaid over Fig. 2. I know it is a bit of a tedious task, but a picture is worth a thousand words!

Line 234, “the cumulative posterior tree space explored”. It is not immediately clear what you mean. Can you express these results in a less technical manner?

Paragraph starting at line 241. It is suggested that the 1.64 Ga biomarker calibration is incompatible with the HGT constraints were Chlorobiaceae is recipient. Then, you use this result to suggest that the biomarkers were not produced by Chlorobiaceae. But it is not clear why this is the case or what is the nature of this incompatibility.

The calibration is a younger bound, implying that crown Chlorobiaceae is older than 1.64 Ga, and that is exactly what I see in Fig. 2, for example. Am I wrong? So, it is not immediately obvious why the calibration is incompatible. Can you elaborate on this, perhaps on the supplements? I would appreciate being able to see in a figure how the specific transfer events are incompatible with the dates produced in the presence of the calibration for this particular case, and then made compatible once the new dates are obtained in the absence of the calibration.

I think you should also briefly mention and discuss the consistency of your best scenario with the Proterozoic fossil record of heterocystous cyanobacteria, the so-called fossil akinetes, and those presented in [4], for example. I think mentioning the heterocystous cyanobacteria fossils is important because the post-GOE “crown cyanobacteria” clocks published always produce an origin of this clade around the Phanerozoic. Lastly, I think you should also highlight the consistency of your data with the fossil record of eukaryotes themselves, since there has been a case made for crown group eukaryotes being younger than ~1.2 Ga, see [5], but see this [6].

Additional comment on HGT. There is a lot of talk in the evolution of photosynthesis about the acquisition of the process via HGT. I am quite puzzled by the fact that your HGT retrieval process did not detect or include any photosynthesis genes. There are well known transfer events of chlorophyll synthesis genes between stem Chlorobi and stem Chloroflexi... unclear direction, but I am puzzled by the fact that none of these made it to your final dataset. It has been suggested that photosynthesis was only acquired by the Chloroflexi about a billion years ago, yet you did not seem to identify any events consistent with that. It has also been suggested that cyanobacteria could have obtained photosynthesis via HGT after the divergence from melainabacteria, and so I would have expected that such events would be detectable through your retrieval methodology. Can you comment on this?

Lines 284 to 288. You say: "Similarly constraining the older-bound estimate on the origin of oxygenic photosynthesis..." As I discussed in the Assumptions section above, here you are making some assumptions that are unproven. You are not measuring or constraining the origin of oxygenic photosynthesis. You are talking here about the node that represents the last common ancestor of melainabacteria and cyanobacteria. The data that you present do not inform on the most likely ancestral metabolic capacity of this node. Please, rephrase for precision.

Lines 318 to 328. This is a bit confusing. In lines, 320 to 325 you say that the transitions towards the origin of oxygenic photosynthesis or the emergence of a photosystem capable of splitting water to oxygen cannot be resolved with a species trees. However, you do seem to be sure that these transitions occurred after the divergence of melainabacteria. How come?

Lines 338-340. Why is it likely that they represent extinct groups? Why is it not more likely that we have not completely described the diversity of cyanobacteria? I give you some examples... the discovery of Gloeomargarita, which might be common in hot springs but completely unknown just a few years ago. Another example, *Aurora vandensis*... then it just turned out that this has a sister lineage that lives in association with hornworts [7]! Who would have thought? Other example outside cyanobacteria, phototrophic Gemmatimonadetes that are indeed quite abundant, or Chloroflexi with type I photosystems. And these are serendipitous discoveries, not being born from projects that were dedicated to the mapping or description of a group's diversity.

Lines 349-351, but also SI text 5. "being early anoxygenic phototrophs of a likely extinct bacterial lineage..." How can you tell the difference between an extinct lineage of anoxygenic phototroph and an extant lineage of non-photosynthetic bacteria whose ancestors have been losing photosynthesis starting over three billion years ago? When gene loss is such a common and constant process. By what method or measurement do you distinguish between those two possibilities? How can you even tell that this is actually likely?

How can you tell that the ancestor represented by the oldest node in Fig. 2 did not have the capacity for photosynthesis when you have phototrophic lineages on all parts of this tree, without showing us or addressing the evolution of the components of photosynthesis? I could argue that the phylogeny of type I photosystems, which features similar distances and rates between Chlorobi and cyanobacteria as you show here for your concatenated sequences, indicate that their last common ancestor had type I photosystems.

In addition, I believe Figure 4 is a bit misleading and misrepresents the evolution of photosynthesis. It gives the impression to the reader that evolution is a linear process. It implies that anoxygenic photosynthesis gradually evolved to become oxygenic, kind of like ferns evolving to become flowering plants, or chimpanzees evolving to become human. In reality, the early evolution of the photosystems, the complexes that enable photosynthesis, is best described as a deep dichotomy, a deep bifurcation leading to those lineages of photosystems used in anoxygenic and those used in oxygenic photosynthesis. These early stages of photosystem evolution cannot be mapped out onto a species tree of bacteria, as photosystem phylogenies suggest that these predate the main bacterial radiation. It is usually assumed [not based on any

particular solid evidence, but on decades-old and weak rationales], that the ancestral states of the earliest photosystem were anoxygenic, but these views have been challenged.

I would like you to compare your data with that of ref. [2]. The same node that in your tree you call “crown bacteria” Jabłońska and Tawfik named “last universal oxygen ancestor”. The authors traced back to this point a major radiation of oxygen-using enzymes. They state this node is 3.1 Ga old based on the data compiled by TimeTree, but it is equivalent to your “crown bacteria”, which you calculate to be about 3.6 Ga old. A similar pattern of evolution was noted by [3], in which they correlate the bacterial expansion of diversity with the availability of oxygen. The point that I want to make with these reflections, is that these bits of the discussion that I highlighted, including figure 4, represent your perception of a scenario that you think likely, based on a series of assumptions that you take for granted, and integrating only selected pieces of evidence you might be more familiar with. I encourage you to reconsider how you present the discussion:

- 1) I think it is important that you discuss previous molecular clock studies on bacteria and cyanobacteria, and relate your findings to that extensive literature, before you relate them to the geochemical or microfossil record. Acknowledge and consider seriously those who have come before you.
- 2) Explicitly acknowledge and state your assumptions.
- 3) Be extremely cautious with the statements of what you perceive to be more or less likely contrasted to a substantiated body of evidence.

Methods. Starting line 444, Application of HGT-based age constraints. This methodology is not clear at all, and I am not sure it would allow others to reproduce your work. You sampled “1k and 10k chronograms”. What is the rationale for this? What is a 1k or 10k chronogram? You mean after that many cycles in phylobayes? Remember that not all software packages work the same way.

You then say that each tree is determined to be compatible or incompatible. How is this done? What is the procedure for this assessment? What does “incompatible” actually means in terms of divergence times?

I am not sure what you mean with “tree space per HGT”. How does the attrition curve combine constraints imposed by multiple HGTs? Do you need to modify the phylobayes algorithm to be able to do this? Do you have to input a new series of calibration files incorporating the HGT information and then recalculate a clock? How do you use the HGT info to recalculate the confidence intervals, for example?

Can you please rework this section?

Line 405, SI Dataset 1. The trees provided labelled Model B and D both included plastid sequences. The trees provided did not appear to have branch support values.

Cited references

1. Oliver, T., P. Sánchez-Baracaldo, A.W. Larkum, A.W. Rutherford, and T. Cardona, Time-resolved comparative molecular evolution of oxygenic photosynthesis. *Biochim. Biophys. Acta*, 2021: 148400. DOI: <https://doi.org/10.1016/j.bbabi.2021.148400>.
2. Jabłońska, J. and D.S. Tawfik, The evolution of oxygen-utilizing enzymes suggests early biosphere oxygenation. *Nat. Ecol. Evol.*, 2021. DOI: 10.1038/s41559-020-01386-9.
3. David, L.A. and E.J. Alm, Rapid evolutionary innovation during an Archaean genetic expansion. *Nature*, 2011. 469: 93-96. DOI: 10.1038/Nature09649.
4. Pang, K., Q. Tang, L. Chen, B. Wan, C. Niu, X. Yuan, and S. Xiao, Nitrogen-fixing heterocystous cyanobacteria in the tonian period. *Curr Biol*, 2018. 28: 616-622 e1. DOI:

10.1016/j.cub.2018.01.008.

5. Porter, S.M., Insights into eukaryogenesis from the fossil record. *Interface Focus*, 2020. DOI: 10.1098/rsfs.2019.0105.

6. Bengtson, S., T. Sallstedt, V. Belivanova, and M. Whitehouse, Three-dimensional preservation of cellular and subcellular structures suggests 1.6 billion-year-old crown-group red algae. *Plos Biol*, 2017. 15: e2000735. DOI: 10.1371/journal.pbio.2000735.

7. Rahmatpour, N., D.A. Hauser, J.M. Nelson, P.Y. Chen, J.C. Villarreal A., M.-Y. Ho, and F.-W. Li, Revisiting the early evolution of Cyanobacteria with a new thylakoid-less and deeply diverged isolate from a hornwort. *bioRxiv*, 2021: 2021.02.18.431691. DOI: 10.1101/2021.02.18.431691.

Review form: Reviewer 2

Recommendation

Major revision is needed (please make suggestions in comments)

Scientific importance: Is the manuscript an original and important contribution to its field?

Acceptable

General interest: Is the paper of sufficient general interest?

Acceptable

Quality of the paper: Is the overall quality of the paper suitable?

Acceptable

Is the length of the paper justified?

Yes

Should the paper be seen by a specialist statistical reviewer?

No

Do you have any concerns about statistical analyses in this paper? If so, please specify them explicitly in your report.

No

It is a condition of publication that authors make their supporting data, code and materials available - either as supplementary material or hosted in an external repository. Please rate, if applicable, the supporting data on the following criteria.

Is it accessible?

Yes

Is it clear?

Yes

Is it adequate?

Yes

Do you have any ethical concerns with this paper?

Yes

Comments to the Author

Fournier et al. combine fossil-based node age calibrations with horizontal gene transfer derived relative node age constraints to date cyanobacteria using a variety of relaxed molecular clock

models.

The question is interesting and the approach applied is novel.

Major Issues:

1. How reliable are the gene trees used in the transfer inference? It seems plausible given the evolutionary time scales in question that the model used for gene tree inference (LG+G) might be inadequate for some, or indeed most, of the gene families. Given the small number of families containing the transfer of interest (<50), it would be of interest to repeat the tree inference aiming for the best fitting model for each gene family (e.g. best-fitting model in IQTree) to establish confidence in the transfers implied by the topology of the trees.

3. Are gene tree branch supports (e.g. bootstrap values) on gene phylogenies taken into account? It might be that groups that appear in an unexpected position are not supported.

3. Are the same transfers (index HGTs) recovered by gene tree - species tree reconciliation methods, in particular ones, such as GeneRax (Benoit 2020 <https://doi.org/10.1093/molbev/msaa141>) that are able to take into account gene tree uncertainty?

4. Removing the two transfers for Chlorobiaceae improves the general agreement with the HGTs but does this really mean that source of error is coming from the calibration in Chlorobiaceae? Could there be another possible explanation? How can we determine if the calibration or the transfers that imply the conflicting relative age constraints are more likely to be correct?

5. The authors find that evolutionary model choice is a major determinant of divergence time estimates, but only investigate different models for the evolutionary rates (UGAM, LN and CIR) and for the divergence time prior (birth-death vs uniform), but do not investigate what effect the choice of substitution model may have. In particular, does using CAT-GTR in the RMC dating step have a substantial effect compared to the C20 mode used? (The number of mixture components in site-heterogenous models can have a strong effect on branch lengths, cf. FIG. 3. in Schrepf et al. 2020 <https://doi.org/10.1093/molbev/msaa145>)

Minor Points:

Abstract: The authors state both the precision and accuracy of dates are improved by incorporating relative age constraints. A reduction in uncertainty is expected, but how do we (or indeed can we) know that accuracy is improved? Do the authors demonstrate an increase in accuracy?

lines 92-94 "species tree reconciliations" gene trees are reconciled with species trees, I assume the authors mean "gene tree - species tree reconciliation methods"?

Decision letter (RSPB-2021-0675.R0)

04-May-2021

Dear Professor Fournier:

Your manuscript has now been peer reviewed and the reviews have been assessed by an Associate Editor. The reviewers' comments (not including confidential comments to the Editor) and the comments from the Associate Editor are included at the end of this email for your

reference. As you will see, the reviewers and the Editors have raised some concerns with your manuscript and we would like to invite you to revise your manuscript to address them.

Research ethics:

Use of animals and field studies:

It is a condition of publication that you make available the data and research materials supporting the results in the article. Please see our Data Sharing Policies (<https://royalsociety.org/journals/authors/author-guidelines/#data>). Datasets should be deposited in an appropriate publicly available repository and details of the associated accession number, link or DOI to the datasets must be included in the Data Accessibility section of the article (<https://royalsociety.org/journals/ethics-policies/data-sharing-mining/>). Reference(s) to datasets should also be included in the reference list of the article with DOIs (where available).

Please submit a copy of your revised paper within three weeks. If we do not hear from you within this time your manuscript will be rejected. If you are unable to meet this deadline please let us know as soon as possible, as we may be able to grant a short extension.

Best wishes,
Professor Gary Carvalho
mailto:proceedingsb@royalsociety.org

Associate Editor
Board Member: 1
Comments to Author:

We have received two highly complementary reviews that are generally favourable, but suggest that significant work should be invested in revising the manuscript.

Reviewer 1 focused on the discussion and interpretations and did not request further analyses. The reviewer is enthusiastic about the paper but offers many specific critiques of the reasoning and wording. There are also several interesting suggestions for adding material to the supplement.

The reviewer starts by pointing out a key conceptual issue, i.e., that key traits and crown groups do not evolve absolutely simultaneously. The reviewer is right that these ideas need to be untangled in every instance where confusion might arise. On a related point, the idea that melainabacteria may have originally been photosynthetic but later lost that ability is suggested, and this may not be difficult to address.

Many of the other suggestions regard matters of phrasing and organisation and are also easily dealt with. For example, ending the introduction with a clearer, less abrupt transition is indeed conventional wisdom. The beginning of the results section is actually very specific on what tests of robustness were carried out, but could use a little elaboration, as suggested.

Queries about the section starting at line 111 point to a structural issue. When the main sections under Results are summarised in the first paragraph (a good idea), it is indicated that evolutionary models will be addressed last. However, the first section starts to discuss

evolutionary models when the reader might not expect it. Some reorganisation of the Results section to make sure the outline is followed is suggested.

The queries involving line 137 (about assumptions regarding the evolution of Eukarya) and the paragraph about "compatibilities" and Chlorobiaceae are particularly significant. Wording about "extinct groups" at line 339 does need to be hedged.

Fig. 4 is particularly important because it summarises key take-home messages and is intelligible to anyone. It is therefore important to clarify in the text why the authors think anoxygenic photosynthesis was present across a grade of cyanobacteria leading up to oxygenic cyanobacteria. Otherwise, it's hard to make a case that the former led into the latter. This matter comes up at lines 320-322, which state that anoxygenic phototrophy was "presumably" intermediate but do not make a clear case. It also comes up in the passage about "early anoxygenic phototrophs" at lines 349-351, where the reviewer suggested an alternative hypothesis.

The reviewer is right to suggest saying exactly what "compatible or incompatible" means at line 447. Additional queries here are also non-trivial.

Reviewer 2 offered a number of queries regarding technical details of the analysis, but did not question the scholarship or interpretations. These queries amount to requesting significant reworking of the analyses (e.g., using different models for different gene families, or substituting CAT-GTR for C20). A revised manuscript should implement some of these suggestions, and if the authors decline to implement them, reasons for not doing so need to be detailed.

Reviewer(s)' Comments to Author:

Referee: 1

Comments to the Author(s)

Fournier et al. present an excellent, comprehensive, and exciting molecular clock analysis tackling the issue of when the last common ancestor of cyanobacteria occurred. The authors enable the use of novel fossil calibrations by sequencing new genomes strategically located in a phylogeny of bacteria. And in addition, they implement an approach to improve the "precision and accuracy" of divergence time estimation by further constraining the clocks with a substantial number of horizontal gene transfer (HGT) events.

The authors conclude that the last common ancestor of cyanobacteria occurred well before the Great Oxidation Event, with the divergence of photosynthetic and non-photosynthetic cyanobacteria occurring between 3.25 and 3.50 Ga in their best preferred model. These observations were well supported by the presented data.

I will point out a few things that should be presented or discussed more adequately. I will also challenge a number of implicit assumptions that are not necessarily supported by the presented data or the current state of knowledge, but that in my opinion, do not take away from the quality of the work and results or the main conclusion regarding timing the last common ancestor of cyanobacteria. I wish my feedback to be taken as an ongoing and engaged discussion and a conversation, as I believe all my points can be reasonably addressed in a round of revisions.

I appreciate the authors making all alignments, trees, dates and rates files available for peer-review. That was a real treat and I enjoyed myself exploring these!

Overall, I think it is a fantastic effort. Thank you.

Key technical concepts

Line 24 and line 51, but see also further below. I think it is important to decouple the concept of "oxygenic photosynthesis" and the concept of "cyanobacteria". They are not the same thing and

their evolution is not coincidental in time. As our understanding of the evolution of photosynthesis advances, it is important that more precision is used to describe what is a nuanced evolutionary process.

There are four key elements here: 1) The last common ancestor of cyanobacteria giving rise to crown-group cyanobacteria, 2) the last common ancestor of melainabacteria (Vampiromicrobia) and cyanobacteria. These two are well defined points in the species tree of bacteria. 3) Stem-group cyanobacteria, which represent ancestors located along the branch connecting the last common ancestors 1) and 2). And 4) the origin of oxygenic photosynthesis, which cannot be precisely located in a species tree.

You mostly use “stem cyanobacteria” to refer to point 2) when you report specific dates, but you also use it as in meaning 3). For example, as in line 318: “Molecular clock analysis does not directly reveal how long stem cyanobacteria were oxygenic before their crown diversification.” Importantly, point 1) and 2) represent a precise point in time, while meaning 3) represents a period of time, a series of undefined ancestors. Thus, as used in 318, it becomes a bit confusing.

Therefore, when you say in the abstract, “these results support oxygenic photosynthesis evolving several hundred million years before the Great Oxygenation Event (GOE)”, what do you actually mean? Do you mean that the results support a last common ancestor of cyanobacteria evolving several hundred million years before the GOE? Or do you mean that your data supports oxygenic photosynthesis evolving at least several hundred million years before the GOE? Given that the mean age for “stem cyanobacteria” under the preferred scenario (CIR nobd, BE+HGT) is a billion years before the GOE... The way it is expressed in line 24 in the abstract, it feels like you are using “oxygenic photosynthesis” and “crown cyanobacteria” as synonymous.

However, in line 51, it feels like the authors are using “oxygenic photosynthesis” and “stem cyanobacteria” as synonymous. Also note that in reference [17] and [18] it is actually “crown cyanobacteria” that are reported to be over a billion years after the GOE! Not just “stem cyanobacteria”.

I would like to encourage you to be a bit more mindful in the way these different concepts are used through the text. I also want to strongly recommend that you explicitly define in the introduction the terminology that you are going to use through the text; do not just assume the reader knows the way you personally like to refer to things. ♦♦ Tweak and refine as to be precise with what you mean, and double check for consistency.

Assumptions

This leads me to an important aspect of the way the data is interpreted. There is an implicit assumption that oxygenic photosynthesis originated after the divergence of melainabacteria. This is by no means a fact. That the last common ancestor of cyanobacteria and melainabacteria was non-photosynthetic is not a fact, it is only a hypothesis. When we bring the evolution of the photosystems into the picture, then it is also quite possible that oxygenic photosynthesis predates what you call “stem cyanobacteria” [1]. A recent analysis of the origin of oxygen-using enzymes is also consistent with photosystem evolution [2], echoing results from nearly a decade ago [3].

I do not think it is necessary for you to discuss these aspects in your paper in any detail. However, I do think you should acknowledge and state your assumption in a direct, clear and explicit way in the introduction, discussion, or where you find it more suitable.

Other comments

Line 53, “Much of this incongruity”. How much? I personally think the incongruity is mostly due to what you said in lines 48 to 50: a combination of factors. Perhaps rephrase?

Line 72, do you mean SI text 2? It is not clear. It would be a nice gesture for the readers if you could perhaps link the narrative of the SI text in a more fluid way to the main text. There is an

abrupt change of pace when you shift from main to SI text, in such a way that I am confused as to whether you sent me to the right place: in particular, SI text 1, 2, and 5. Just add an introductory or linking opening statement. Thank you.

It would be nice too if you give the SI texts titles that reflect the content of the text.

End of introduction and start of results section. As a matter of convention, it is helpful to have a little paragraph at the end of the introduction that links to the results section, telling in a few brief statements what you did, your rationale, and/or what you found. This is particularly important given that the introduction is not followed by the methodology.

The results section also opens in a way that it is not clear what is the question that you are going to address, what is the system that you are working with, or the hypothesis that you will be testing. As it is now, it is just really hard to penetrate and you will have the reader jumping all over the place trying to deduce or guess what you mean or did.

Line 116, the first data you refer to in the text is SI Table 1. When I went to find that table, I found DD, BH, +older bounds, bd, no bd. At this point none of these have been introduced to the reader. I am left then trying to find what these are: it took me a while to find out that the key with the experiments is found in SI Table 3, which also use some code of white, grey, filled boxes with numbers or an x, with no explanation.

I encourage the authors to reconsider how the data is presented, so that it makes for a smoother reading. To begin with, it is better if the first data that you choose to show is in the main text and not in the supplements. Perhaps you can structure things such that you first direct the reader to Table 1.

Your first paragraph of proper results, starting in line 111, is also a bit messy and hard to disentangle. Consider presenting the data first, then presenting the interpretation. For example, "under uncorrelated branch rate models the ages are X and Y. In contrast, under autocorrelated models the ages are A and B. We think these differences are caused by factors M and N, as proposed in reference [16]."

Line 120-123. It is stated that uncorrelated models independently assign higher penalties across branches inferred to be evolving at faster rates. What do you mean with "independently"? Is this something that you know from how the algorithms were developed or is this your rationalization of the observed effects? Can you provide a reference where these technical aspects of rate models are presented?

One major difference between uncorrelated and autocorrelated is that the former assumes an average rate for each internal node, while in the latter, rates at the internal nodes can vary to any magnitude. Is that linked to the penalties you talk about? Perhaps you want to keep it simple and elaborate on this on a supplementary text for the more specialist audience. Consider that the broader readership of Proceedings B interested in the evolution of cyanobacteria and photosynthesis may not be aware of what autocorrelated, uncorrelated, and birth-death means.

You use the expression, "far more robust", what is your measurement for robustness? How much robust exactly is "far more"?

You conclude in the same paragraph stating that autocorrelated models are more consistent and reliable than uncorrelated. When looking at the patterns of rates of evolution from the datasets provided, I noted a substantial decrease in the rates of evolution with decreasing geological time, and a zone of large scatter after the GOE. I was not able to associate the rates to taxa, given that a file with labelled tree nodes was not provided (can you make these also available?), but I wonder if you could describe and discuss the patterns of rate change that you observe in your preferred scenarios as a supplementary text. I would appreciate that.

Section “Impact of eukaryotic calibrations”. I think the data and interpretation presented in this section can be better understood in terms of effects on the rates of evolution, rather than effects on divergence times. It is known that eukaryotic sequences evolve relatively faster than those in cyanobacteria, as clearly seen in the trees you provided in Dataset S1: the “long branches within plastid lineages”. When you add or remove taxa and calibrations as you describe, do these translate into increases or decreases on the rates of evolution?

For example, lines 141 to 150. In the absence of cyanobacterial calibrations and constraining photosynthetic eukaryotes to younger than 1.8 Ga, it is expected that this will translate into an acceleration of the rates of evolution for cyanobacteria, because the calculation of the rates will be dependent on the “younger than 1.8 Ga calibration”, which is allocated to the faster evolving eukaryotic sequences embedded within cyanobacteria. Therefore, it is expected that the measured ages for crown Cyanobacteria will appear younger given the calculated faster rates of amino acid substitutions.

Consider rephrasing for clarity.

Line 137. “Far more robust”. You make it sound like timing the evolution of eukaryotes with molecular clock is a done deal, or that the fossil record of eukaryotes is well understood. I am sure you recognize that molecular clocks have also generated massive uncertainties of nearly a billion years for crown eukaryotes. And that there are a bunch of eukaryote-like fossils and trace fossils that are older than 2.0 Ga. Perhaps you want to be a bit more cautious about how you make an interpretation of the utility of including plastid lineages in this type of analysis.

In any case, it is not clear what the take-home message of this section is regarding timing the evolution of cyanobacteria. Perhaps you wish to restate or rephrase for additional impact.

Line 152, Section (2). I think enabling these new calibrations is amazing. I wonder if you have images of the microbial mats where the MAGs came from, or microscopy images of the morphological analogues of *Eoentophysalis*, or the tightly coiled cyanobacteria strains? It would be great if you could share some of these in the supplementary materials.

Section starting in line 224. I want to say that I am indeed very positively impressed by the HGT-enhanced clock results and the effect the approach has on narrowing the confidence intervals.

I think it would be very useful to the reader if you can provide in a separate supplementary figure, the HGT transfer events that were studied, overlaid over Fig. 2. I know it is a bit of a tedious task, but a picture is worth a thousand words!

Line 234, “the cumulative posterior tree space explored”. It is not immediately clear what you mean. Can you express these results in a less technical manner?

Paragraph starting at line 241. It is suggested that the 1.64 Ga biomarker calibration is incompatible with the HGT constraints were Chlorobiaceae is recipient. Then, you use this result to suggest that the biomarkers were not produced by Chlorobiaceae. But it is not clear why this is the case or what is the nature of this incompatibility.

The calibration is a younger bound, implying that crown Chlorobiaceae is older than 1.64 Ga, and that is exactly what I see in Fig. 2, for example. Am I wrong? So, it is not immediately obvious why the calibration is incompatible. Can you elaborate on this, perhaps on the supplements? I would appreciate being able to see in a figure how the specific transfer events are incompatible with the dates produced in the presence of the calibration for this particular case, and then made compatible once the new dates are obtained in the absence of the calibration.

I think you should also briefly mention and discuss the consistency of your best scenario with the Proterozoic fossil record of heterocystous cyanobacteria, the so-called fossil akinetes, and those presented in [4], for example. I think mentioning the heterocystous cyanobacteria fossils is important because the post-GOE “crown cyanobacteria” clocks published always produce an origin of this clade around the Phanerozoic. Lastly, I think you should also highlight the consistency of your data with the fossil record of eukaryotes themselves, since there has been a case made for crown group eukaryotes being younger than ~1.2 Ga, see [5], but see this [6].

Additional comment on HGT. There is a lot of talk in the evolution of photosynthesis about the acquisition of the process via HGT. I am quite puzzled by the fact that your HGT retrieval process did not detect or include any photosynthesis genes. There are well known transfer events of chlorophyll synthesis genes between stem Chlorobi and stem Chloroflexi... unclear direction, but I am puzzled by the fact that none of these made it to your final dataset. It has been suggested that photosynthesis was only acquired by the Chloroflexi about a billion years ago, yet you did not seem to identify any events consistent with that. It has also been suggested that cyanobacteria could have obtained photosynthesis via HGT after the divergence from melainabacteria, and so I would have expected that such events would be detectable through your retrieval methodology. Can you comment on this?

Lines 284 to 288. You say: “Similarly constraining the older-bound estimate on the origin of oxygenic photosynthesis...” As I discussed in the Assumptions section above, here you are making some assumptions that are unproven. You are not measuring or constraining the origin of oxygenic photosynthesis. You are talking here about the node that represents the last common ancestor of melainabacteria and cyanobacteria. The data that you present do not inform on the most likely ancestral metabolic capacity of this node. Please, rephrase for precision.

Lines 318 to 328. This is a bit confusing. In lines, 320 to 325 you say that the transitions towards the origin of oxygenic photosynthesis or the emergence of a photosystem capable of splitting water to oxygen cannot be resolved with a species trees. However, you do seem to be sure that these transitions occurred after the divergence of melainabacteria. How come?

Lines 338-340. Why is it likely that they represent extinct groups? Why is it not more likely that we have not completely described the diversity of cyanobacteria? I give you some examples... the discovery of Gloeomargarita, which might be common in hot springs but completely unknown just a few years ago. Another example, *Aurora vandensis*... then it just turned out that this has a sister lineage that lives in association with hornworts [7]! Who would have thought? Other example outside cyanobacteria, phototrophic Gemmatimonadetes that are indeed quite abundant, or Chloroflexi with type I photosystems. And these are serendipitous discoveries, not being born from projects that were dedicated to the mapping or description of a group's diversity.

Lines 349-351, but also SI text 5. “being early anoxygenic phototrophs of a likely extinct bacterial lineage...” How can you tell the difference between an extinct lineage of anoxygenic phototroph and an extant lineage of non-photosynthetic bacteria whose ancestors have been losing photosynthesis starting over three billion years ago? When gene loss is such a common and constant process. By what method or measurement do you distinguish between those two possibilities? How can you even tell that this is actually likely?

How can you tell that the ancestor represented by the oldest node in Fig. 2 did not have the capacity for photosynthesis when you have phototrophic lineages on all parts of this tree, without showing us or addressing the evolution of the components of photosynthesis? I could argue that the phylogeny of type I photosystems, which features similar distances and rates between Chlorobi and cyanobacteria as you show here for your concatenated sequences, indicate that their last common ancestor had type I photosystems.

In addition, I believe Figure 4 is a bit misleading and misrepresents the evolution of photosynthesis. It gives the impression to the reader that evolution is a linear process. It implies that anoxygenic photosynthesis gradually evolved to become oxygenic, kind of like ferns evolving to become flowering plants, or chimpanzees evolving to become human. In reality, the early evolution of the photosystems, the complexes that enable photosynthesis, is best described as a deep dichotomy, a deep bifurcation leading to those lineages of photosystems used in anoxygenic and those used in oxygenic photosynthesis. These early stages of photosystem evolution cannot be mapped out onto a species tree of bacteria, as photosystem phylogenies suggest that these predate the main bacterial radiation. It is usually assumed [not based on any particular solid evidence, but on decades-old and weak rationales], that the ancestral states of the earliest photosystem were anoxygenic, but these views have been challenged.

I would like you to compare your data with that of ref. [2]. The same node that in your tree you call “crown bacteria” Jabłońska and Tawfik named “last universal oxygen ancestor”. The authors traced back to this point a major radiation of oxygen-using enzymes. They state this node is 3.1 Ga old based on the data compiled by TimeTree, but it is equivalent to your “crown bacteria”, which you calculate to be about 3.6 Ga old. A similar pattern of evolution was noted by [3], in which they correlate the bacterial expansion of diversity with the availability of oxygen.

The point that I want to make with these reflections, is that these bits of the discussion that I highlighted, including figure 4, represent your perception of a scenario that you think likely, based on a series of assumptions that you take for granted, and integrating only selected pieces of evidence you might be more familiar with. I encourage you to reconsider how you present the discussion:

- 1) I think it is important that you discuss previous molecular clock studies on bacteria and cyanobacteria, and relate your findings to that extensive literature, before you relate them to the geochemical or microfossil record. Acknowledge and consider seriously those who have come before you.
- 2) Explicitly acknowledge and state your assumptions.
- 3) Be extremely cautious with the statements of what you perceive to be more or less likely contrasted to a substantiated body of evidence.

Methods. Starting line 444, Application of HGT-based age constraints. This methodology is not clear at all, and I am not sure it would allow others to reproduce your work. You sampled “1k and 10k chronograms”. What is the rationale for this? What is a 1k or 10k chronogram? You mean after that many cycles in phylobayes? Remember that not all software packages work the same way.

You then say that each tree is determined to be compatible or incompatible. How is this done? What is the procedure for this assessment? What does “incompatible” actually means in terms of divergence times?

I am not sure what you mean with “tree space per HGT”. How does the attrition curve combine constraints imposed by multiple HGTs? Do you need to modify the phylobayes algorithm to be able to do this? Do you have to input a new series of calibration files incorporating the HGT information and then recalculate a clock? How do you use the HGT info to recalculate the confidence intervals, for example?

Can you please rework this section?

Line 405, SI Dataset 1. The trees provided labelled Model B and D both included plastid sequences. The trees provided did not appear to have branch support values.

Cited references

1. Oliver, T., P. Sánchez-Baracaldo, A.W. Larkum, A.W. Rutherford, and T. Cardona, Time-resolved comparative molecular evolution of oxygenic photosynthesis. *Biochim. Biophys. Acta*, 2021: 148400. DOI: <https://doi.org/10.1016/j.bbabi.2021.148400>.
2. Jabłońska, J. and D.S. Tawfik, The evolution of oxygen-utilizing enzymes suggests early biosphere oxygenation. *Nat. Ecol. Evol.*, 2021. DOI: 10.1038/s41559-020-01386-9.
3. David, L.A. and E.J. Alm, Rapid evolutionary innovation during an Archaean genetic expansion. *Nature*, 2011. 469: 93-96. DOI: 10.1038/Nature09649.
4. Pang, K., Q. Tang, L. Chen, B. Wan, C. Niu, X. Yuan, and S. Xiao, Nitrogen-fixing heterocystous cyanobacteria in the tonian period. *Curr Biol*, 2018. 28: 616-622 e1. DOI: 10.1016/j.cub.2018.01.008.
5. Porter, S.M., Insights into eukaryogenesis from the fossil record. *Interface Focus*, 2020. 10. DOI: 10.1098/rsfs.2019.0105.
6. Bengtson, S., T. Sallstedt, V. Belivanova, and M. Whitehouse, Three-dimensional preservation of cellular and subcellular structures suggests 1.6 billion-year-old crown-group red algae. *Plos Biol*, 2017. 15: e2000735. DOI: 10.1371/journal.pbio.2000735.
7. Rahmatpour, N., D.A. Hauser, J.M. Nelson, P.Y. Chen, J.C. Villarreal A., M.-Y. Ho, and F.-W. Li, Revisiting the early evolution of Cyanobacteria with a new thylakoid-less and deeply diverged isolate from a hornwort. *bioRxiv*, 2021: 2021.02.18.431691. DOI: 10.1101/2021.02.18.431691.

Referee: 2

Comments to the Author(s)

Fournier et al. combine fossil-based node age calibrations with horizontal gene transfer derived relative node age constraints to date cyanobacteria using a variety of relaxed molecular clock models.

The question is interesting and the approach applied is novel.

Mayor Issues:

1. How reliable are the gene trees used in the transfer inference? It seems plausible given the evolutionary time scales in question that the model used for gene tree inference (LG+G) might be inadequate for some, or indeed most, of the gene families. Given the small number of families containing the transfer of interest (<50), it would be of interest to repeat the tree inference aiming for the best fitting model for each gene family (e.g. best-fitting model in IQTree) to establish confidence in the transfers implied by the topology of the trees.
3. Are gene tree branch supports (e.g. bootstrap values) on gene phylogenies taken into account? It might be that groups that appear in an unexpected position are not supported.
3. Are the same transfers (index HGTs) recovered by gene tree - species tree reconciliation methods, in particular ones, such as GeneRax (Benoit 2020 <https://doi.org/10.1093/molbev/msaa141>) that are able to take into account gene tree uncertainty?
4. Removing the two transfers for Chlorobiaceae improves the general agreement with the HGTs but does this really mean that source of error is coming from the calibration in Chlorobiaceae? Could there be another possible explanation? How can we determine if the calibration or the transfers that imply the conflicting relative age constraints are more likely to be correct?
5. The authors find that evolutionary model choice is a major determinant of divergence time estimates, but only investigate different models for the evolutionary rates (UGAM, LN and CIR) and for the divergence time prior (birth-death vs uniform), but do not investigate what effect the choice of substitution model may have. In particular, does using CAT-GTR in the RMC dating step have a substantial effect compared to the C20 mode used? (The number of mixture

components in site-heterogenous models can have a strong effect on branch lengths, cf. FIG. 3. in Schrempf et al. 2020 <https://doi.org/10.1093/molbev/msaa145>)

Minor Points:

Abstract: The authors state both the precision and accuracy of dates are improved by incorporating relative age constraints. A reduction in uncertainty is expected, but how do we (or indeed can we) know that accuracy is improved? Do the authors demonstrate an increase in accuracy?

lines 92-94 “species tree reconciliations” gene trees are reconciled with species trees, I assume the authors mean “gene tree - species tree reconciliation methods”?

Author's Response to Decision Letter for (RSPB-2021-0675.R0)

See Appendix A.

RSPB-2021-0675.R1 (Revision)

Review form: Reviewer 1 (Tanai Cardona)

Recommendation

Accept as is

Scientific importance: Is the manuscript an original and important contribution to its field?

Excellent

General interest: Is the paper of sufficient general interest?

Excellent

Quality of the paper: Is the overall quality of the paper suitable?

Excellent

Is the length of the paper justified?

Yes

Should the paper be seen by a specialist statistical reviewer?

No

Do you have any concerns about statistical analyses in this paper? If so, please specify them explicitly in your report.

No

It is a condition of publication that authors make their supporting data, code and materials available - either as supplementary material or hosted in an external repository. Please rate, if applicable, the supporting data on the following criteria.

Is it accessible?

Yes

Is it clear?

Yes

Is it adequate?

Yes

Do you have any ethical concerns with this paper?

No

Comments to the Author

Thank you for taking the time to consider my comments and suggestions in detail. I think these have been addressed satisfactorily. I have no additional feedback. Looking forward to seeing the final version of the manuscript published.

Review form: Reviewer 2

Recommendation

Accept as is

Scientific importance: Is the manuscript an original and important contribution to its field?

Good

General interest: Is the paper of sufficient general interest?

Good

Quality of the paper: Is the overall quality of the paper suitable?

Good

Is the length of the paper justified?

Yes

Should the paper be seen by a specialist statistical reviewer?

No

Do you have any concerns about statistical analyses in this paper? If so, please specify them explicitly in your report.

No

It is a condition of publication that authors make their supporting data, code and materials available - either as supplementary material or hosted in an external repository. Please rate, if applicable, the supporting data on the following criteria.

Is it accessible?

Yes

Is it clear?

Yes

Is it adequate?

Yes

Do you have any ethical concerns with this paper?

No

Comments to the Author

The authors have adequately addressed my concerns.

Decision letter (RSPB-2021-0675.R1)

06-Sep-2021

Dear Professor Fournier

I am pleased to inform you that your manuscript entitled "The Archean Origin of Oxygenic Photosynthesis and Extant Cyanobacterial Lineages" has been accepted for publication in Proceedings B.

Data Accessibility section

Open Access

You are invited to opt for Open Access, making your freely available to all as soon as it is ready for publication under a CCBY licence. Our article processing charge for Open Access is £1700. Corresponding authors from member institutions (<http://royalsocietypublishing.org/site/librarians/allmembers.xhtml>) receive a 25% discount to these charges. For more information please visit <http://royalsocietypublishing.org/open-access>.

Paper charges

Sincerely,
Professor Gary Carvalho
Editor, Proceedings B
mailto: proceedingsb@royalsociety.org

Appendix A

Associate Editor

Board Member: 1

Comments to Author:

We have received two highly complementary reviews that are generally favourable, but suggest that significant work should be invested in revising the manuscript.

We thank the reviewers and editor for their constructive and valuable suggestions. We have endeavored to respond in full to each reviewer, including the addition of substantive new analyses as suggested by reviewer #2.

Reviewer 1 focused on the discussion and interpretations and did not request further analyses. The reviewer is enthusiastic about the paper but offers many specific critiques of the reasoning and wording. There are also several interesting suggestions for adding material to the supplement.

The reviewer starts by pointing out a key conceptual issue, i.e., that key traits and crown groups do not evolve absolutely simultaneously. **The reviewer is right that these ideas need to be untangled in every instance where confusion might arise.** On a related point, the idea that melainabacteria may have originally been photosynthetic but later lost that ability is suggested, and this may not be difficult to address.

We have standardized and clarified our usage of cladistics terms throughout the manuscript, to avoid any confusion or misinterpretation. We also now acknowledge this alternative hypothesis (that oxygenic photosynthesis is an ancestral trait, with subsequent losses in other bacterial lineages, including Melainabacteria). It is clear from Reviewer #1's comments that they favor this scenario, as they specifically mention it in several of the review comments they provided; we would like to point out to the editor that this is not a widely-held view within the field. We have endeavored to provide fair, good-faith responses to these comments, and have updated the manuscript in several places to highlight that the scenario we propose (i.e., the scenario in Figure 4), is predicated on the hypothesis that oxygenic photosynthesis arose within stem Cyanobacteria (which we believe to be the most parsimonious explanation, a position held by most others within the field). This is primarily accomplished through adding the following paragraph:

Lines 296-308:

“These molecular clock analyses do not directly reveal how long cyanobacteria were oxygenic before their crown diversification, because species trees do not currently resolve the evolutionary transitions between intermediate physiological states such as the integration of two photosystems with chlorophyll, or the incorporation of an oxygen-evolving complex. Any groups containing these intermediate sets of characters are either extinct, or remain undiscovered. Under the additional and most parsimonious hypothesis that oxygenic photosynthesis originally arose in the cyanobacterial stem lineage following divergence from a nonphotosynthetic sister group (specific scenarios are discussed in [53][54][55]), the age of total-group Cyanobacteria serves as an older-bound on the origin of oxygenic photosynthesis. Alternatively, studies of individual photosystem genes have led to other, more complex scenarios, that hypothesize oxygenic phototrophy as more ancestral within bacterial diversity[21].”

And subsequently within Figure 4, now titled “Proposed narrative for the cyanobacterial context of Earth's oxidation”.

We remain concerned that the reviewer will disagree with this narrative; in such a case, we wish to assert that we feel it is fully reasonable to propose our own favored scenario informed by our results in a primary research manuscript, even if alternative perspectives exist.

Many of the other suggestions regard matters of phrasing and organisation and are also easily dealt with. For example, **ending the introduction with a clearer, less abrupt transition is indeed conventional wisdom. The beginning of the results section is actually very specific on what tests of robustness were carried out, but could use a little elaboration, as suggested.**

We have revised phrasing and organization throughout, per the suggestions of the reviewers. We have added a transitional paragraph to the end of the introduction:

Lines 97-104:

“Here, we present molecular clock analyses that test the impact of different sets of fossil calibrations on species tree divergence time estimates, including the inclusion of plastid lineages and different evolutionary rate models. We also demonstrate how the use of HGTs can improve model selection criteria and impose more stringent constraints on posterior subsampling for divergence time estimates. These analyses provide consistent support for an Archean origin of crown Cyanobacteria, and a long history of Archean biogenic oxygen production, as well as more precise age estimates for other key divergences in bacterial evolutionary history.”

We also now more specifically enumerate all 4 test conditions that we examine in the beginning of the introduction.

Queries about the section starting at line 111 point to a structural issue. When the main sections under Results are summarised in the first paragraph (a good idea), **it is indicated that evolutionary models will be addressed last. However, the first section starts to discuss evolutionary models when the reader might not expect it. Some reorganisation of the Results section to make sure the outline is followed is suggested.**

We have revised the first section of the Results section titled “(1) Including plastid lineages and calibrations substantially impact cyanobacterial age estimates” to be more clear, as requested by Reviewer #1. In doing so, this section now discusses the impact of these eukaryotic lineages and calibrations before specifically mentioning different evolutionary models. However, different models are unavoidably still discussed in this context, since none of these test conditions are independent from one another. As currently written, the results section addresses, in turn, the impact of calibrations and lineage inclusion across models, then, the impact of the models themselves, and finally, evaluating which models are best (i.e., most consistent with the HGTs), the key age results for these models, and how HGT constraints further increase the precision of these results. We feel it is most important that the section evaluating models follows the section which discusses the age impact of model choice, as models are evaluated using a subset of calibration schemas, as justified by (1) and (2). In order to avoid reader confusion, we now qualify the results outline: “(1) impact of the inclusion of plastid lineages and calibrations across evolutionary models”. Furthermore, this structural change now permits results shown in the main text to be referenced before results provided in the supporting information. We also condensed (4) and the following section, to specifically

discuss and highlight the results for the preferred evolutionary model identified by HGT-based methods, and how additional HGT-constrained sampling further increases the precision of these ages, to make the justification for these favored age estimates more clear.

The queries involving line 137 (about assumptions regarding the evolution of Eukarya) and the paragraph about "compatibilities" and Chlorobiaceae are particularly significant. Wording about "extinct groups" at line 339 does need to be hedged.

We have revised our presentation of the Chlorobiaceae calibrations/age estimates to more clearly justify our interpretation, per the suggestion of the reviewer:

Lines 232-243:

“Only two HGTs showed zero or near-zero (<1.2%) compatibility across all models, both with crown Chlorobiaceae as the recipient clade. Following a previous study [19], we constrained the age of crown Chlorobiaceae with a younger-bound calibration based on 1.64 Ga isorenieratane carotenoid derivatives, as these lipids synthesized by a subset of extant members of this group[48] (SI Table 2). Because this inconsistency was closely associated with one of the few calibrated nodes within our analysis, we questioned the validity of this calibration. Removing the calibration from the divergence time estimation resulted in posterior ages with far greater compatibilities across HGTs where Chlorobiaceae is the recipient (SI Figure 5). This suggests that the calibration is phylogenetically misplaced: the ability to synthesize the observed 1.64 Ga biomarkers could have arisen earlier, either in stem Chlorobiaceae or in an entirely different clade of bacteria [49].”

In discussing “extinct groups”, where this phrase appears, we now further qualify that these groups are either extinct or remain undiscovered.

Fig. 4 is particularly important because it summarises key take-home messages and is intelligible to anyone. **It is therefore important to clarify in the text why the authors think anoxygenic photosynthesis was present across a grade of cyanobacteria leading up to oxygenic cyanobacteria.** Otherwise, it's hard to make a case that the former led into the latter. **This matter comes up at lines 320-322, which state that anoxygenic phototrophy was "presumably" intermediate but do not make a clear case. It also comes up in the passage about "early anoxygenic phototrophs" at lines 349-351, where the reviewer suggested an alternative hypothesis.**

We now provide further justification for this hypothesis in both the text and within Figure 4 itself. We have added a paragraph in the text clarifying that this proposed scenario is predicated on the hypothesis that oxygenic photosynthesis arose within stem Cyanobacteria, and further explain this reasoning, referencing three papers that explore these scenarios in detail. We also now acknowledge that alternative hypothesis exist. (see our response to the question above citing Lines 296-308).

The reviewer is right to suggest saying exactly what "compatible or incompatible" means at line 447. Additional queries here are also non-trivial.

We have addressed these queries and revised the methods text accordingly throughout, specifically:

Lines 418-431:

“Application of HGT-based age constraints

HGT-constrained age estimates were generated for the BE and BH calibration schemas. As provided by the “datedist” output file from PhyloBayes, chronograms sampled after burn-in each represent a likely hypothesis under the given evolutionary model. For each HGT constraint, “compatible” chronograms show the donor being older than the recipient; the percentage of compatible chronograms was then calculated for each HGT constraint (SI Figures 3 and 4). To estimate a posterior age distribution for chronograms compatible with as many HGT constraints as possible, sets of trees passing n HGT constraints were compiled, from $n=1$ to $n=24$ (SI Figure 6). Posterior age estimates were then calculated from the $n=21$ set of trees (Figures 2 and 3). This used the largest number of HGT constraints that recovered a set of compatible chronograms large enough ($n=28$) to estimate age distributions. This method is robust against the impact of a single HGT constraint or biases arising from the stepwise addition of constraints given that each set of n constraints can include any combination of HGTs.”

And within the expanded methods section of the SI:

“SI Text 4: Detailed HGT detection methodology

Bacterial COGs with protein to genome ratios less than 3:1 were used as the starting protein family dataset, to avoid spurious inferences from gene families with histories of extensive gene duplication. For each of the 10,453 protein families examined, patristic distances between protein sequences were transformed to network weights ($e^{-D_{ij}}$, where D_{ij} is the sum of branch lengths between homolog i and homolog j). The Louvain method for community detection[40] was applied to this network, dividing each set of homologs into closely related sub-groups for further analysis. For each phylum, distances between its members and the five closest sequences from each of the other phyla were calculated. Alphaproteobacteria were not included in HGT detection, as the large number of representatives in this dataset proved computationally intractable. Phyla represented by fewer than 5 taxa within a given sub-group were not examined. Inter-phyla distances were deemed to be potentially indicative of HGT events if the median distance to the closest phylum was significantly lower than that of the second-closest phylum ($p=0.01$), resulting in 3,949 sub-groups containing potential HGT candidates. For each sub-group identified as potentially containing an inter-phylum HGT, sequences were re-aligned using MAFFT[41] with automatic parameter selection. Trees were subsequently reconstructed using IQTree[42], under the LG+G model, and supports provided by rapid bootstrapping ($n=1000$) and an approximate likelihood ratio test ($n=1000$). Reconstructed sub-group trees were automatically rooted using Minimal Ancestor Deviation (MAD)[43] (Tria et al., 2017). 249 gene families with nested (paraphyletic) pairs of phyla indicative of index HGTs were then automatically detected using in-house scripts based on the ETE3 python library[44]. Manual vetting of these index HGT candidates was then performed, identifying 38 index HGT candidates. These candidate HGTs were further evaluated using taxon samplings that only included the local topology containing the candidate HGT (donor, recipient, and outgroup sequences), which further restricted the set of inferred index HGTs to 34 events. This local reconstruction ensures that potentially divergent sequences elsewhere in the often-large COG subcluster trees do not have an inordinate influence on the alignment of sequences relevant to the HGT hypothesis being tested, and that the best-fitting model reflects the likelihood of substitutions directly associated with the bipartitions relevant to the HGT inference. For these taxa, sequences were realigned and trees reconstructed in IQtree using the best-fitting evolutionary model. Supports for inferred HGTs were estimated based on the “degenerate” bootstrap support for the nestedness of donor-recipient pairs, as multiple bipartitions

representing specific donor-recipient hypotheses may each be consistent with the relative temporal ordering of the donor and recipient nodes. These values were extracted from bipartition tables (SI Dataset 6) and reported in SI Table 4. Specific narratives for each HGT inference are included in SI Text 5.”

Per one specific comment by Reviewer #1 about comparing to previously published molecular clock results, we wish to further clarify our argument to the editor. We do not feel that primary research results should only be presented accompanied by a review of all previous approaches and results within the field, or that application of results to broader areas of knowledge (in this case, the geochemical record) should only be done once such a meta-analysis has arrived at a consensus. This is a burdensome expectation that hinders the communication and application of research across fields. It is axiomatic that different sets of calibrations, models, and datasets will produce differing age estimates; the key relevance is not how these estimates compare to one another, but how well these calibrations, models, and datasets are justified in their use. Here, we have focused on the latter, and wished to bring our interdisciplinary understanding to interrogate these scientific questions, rather than just performing methods comparisons.

Reviewer 2 offered a number of queries regarding technical details of the analysis, but did not question the scholarship or interpretations. These queries amount to requesting significant reworking of the analyses (e.g., using different models for different gene families, or substituting CAT-GTR for C20). A revised manuscript should implement some of these suggestions, and **if the authors decline to implement them, reasons for not doing so need to be detailed.**

We have implemented Reviewer #2’s suggestions for (1) and (2), adding a substantially more detailed analysis of the gene trees supporting the HGT age constraints. The results of these new analyses were integrated into new versions of relevant figures and tables, and reported in the text accordingly. The data generated supporting these results is now presented in a much-expanded SI Dataset submission on Dryad. We have responded in detail to the other requests, clearly justifying our reasoning for not doing them.

Reviewer(s)' Comments to Author:

Referee: 1

Comments to the Author(s)

Fournier et al. present an excellent, comprehensive, and exciting molecular clock analysis tackling the issue of when the last common ancestor of cyanobacteria occurred. The authors enable the use of novel fossil calibrations by sequencing new genomes strategically located in a phylogeny of bacteria. And in addition, they implement an approach to improve the “precision and accuracy” of divergence time estimation by further constraining the clocks with a substantial number of horizontal gene transfer (HGT) events.

We thank the reviewer for their recognition of our efforts, accurately and succinctly described by them above.

The authors conclude that the last common ancestor of cyanobacteria occurred well before the Great Oxidation Event, with the divergence of photosynthetic and non-photosynthetic cyanobacteria occurring between 3.25 and 3.50 Ga in their best preferred model. These observations were well supported by the presented data.

I will point out a few things that should be presented or discussed more adequately. I will also challenge a number of implicit assumptions that are not necessarily supported by the presented data or the current state of knowledge, but that in my opinion, do not take away from the quality of the work and results or the main conclusion regarding timing the last common ancestor of cyanobacteria. I wish my feedback to be taken as an ongoing and engaged discussion and a conversation, as I believe all my points can be reasonably addressed in a round of revisions.

I appreciate the authors making all alignments, trees, dates and rates files available for peer-review. That was a real treat and I enjoyed myself exploring these!

Overall, I think it is a fantastic effort. Thank you.

Key technical concepts

Line 24 and line 51, but see also further below. I think it is important to decouple the concept of “oxygenic photosynthesis” and the concept of “cyanobacteria”. They are not the same thing and their evolution is not coincidental in time. As our understanding of the evolution of photosynthesis advances, it is important that more precision is used to describe what is a nuanced evolutionary process.

We agree with the reviewer that these things are not coincidental in time. We hope that the subsequent revisions have improved the clarity on this point.

There are four key elements here: 1) The last common ancestor of cyanobacteria giving rise to crown-group cyanobacteria, 2) the last common ancestor of melainobacteria (Vampirovibrionia) and cyanobacteria. These two are well defined points in the species tree of bacteria. 3) Stem-group cyanobacteria, which represent ancestors located along the branch connecting the last common ancestors 1) and 2). And 4) the origin of oxygenic photosynthesis, which cannot be precisely located in a species tree.

We agree with the reviewer that these are the 4 key elements, and we have changed the text throughout to ensure a more precise use of these definitions where they arise.

You mostly use “stem cyanobacteria” to refer to point 2) when you report specific dates, but you also use it as in meaning 3). For example, as in line 318: “Molecular clock analysis does not directly reveal how long stem cyanobacteria were oxygenic before their crown diversification.” Importantly, point 1) and 2) represent a precise point in time, while meaning 3) represents a period of time, a series of undefined ancestors. Thus, as used in 318, it becomes a bit confusing.

We have corrected this possible ambiguity, and now refer to the date of divergence of stem cyanobacteria from the non-photosynthetic outgroup as the age of “total group cyanobacteria” throughout the manuscript. This succinctly and accurately describes the cyanobacteria group, inclusive of all lineages more closely related to Cyanobacteria than the outgroup, descended from element (2) as described by the reviewer above; it is also,

in itself, agnostic about the evolution of character states, which require additional qualifications and interpretation.

Therefore, when you say in the abstract, “these results support oxygenic photosynthesis evolving several hundred million years before the Great Oxygenation Event (GOE)”, what do you actually mean? Do you mean that the results support a last common ancestor of cyanobacteria evolving several hundred million years before the GOE? Or do you mean that your data supports oxygenic photosynthesis evolving at least several hundred million years before the GOE?

We have attempted to further clarify this statement, to avoid any ambiguity:

*“These results support oxygenic photosynthesis evolving **at least** several hundred years before the Great Oxygenation Event (GOE),”*

Given that the mean age for “stem cyanobacteria” under the preferred scenario (CIR nobd, BE+HGT) is a billion years before the GOE... The way it is expressed in line 24 in the abstract, it feels like you are using “oxygenic photosynthesis” and “crown cyanobacteria” as synonymous.

The above revision should preclude this possible interpretation.

However, in line 51, it feels like the authors are using “oxygenic photosynthesis” and “stem cyanobacteria” as synonymous. Also note that in reference [17] and [18] it is actually “crown cyanobacteria” that are reported to be over a billion years after the GOE! Not just “stem cyanobacteria”.

Thank you for pointing out this vagueness; as above, we are referring to bounds here, we have clarified:

Lines 47-54:

“The age of crown Cyanobacteria places a younger-bound constraint on the origin of oxygenic photosynthesis, and is especially important for establishing narratives of planetary biogeochemical change. Due to the sensitivity of molecular clock analyses to sequence datasets, models, and calibrations, and the complex and often subjective criteria for selecting these, age estimates for crown Cyanobacteria have varied widely. Species-tree molecular clock analyses have produced ages that predate the GOE by more than 1 billion years (e.g., [17][18]), less than 0.5 billion years ([19][10][20][21]), very close to the GOE[22][21] or even after the GOE[22][13][20].”

I would like to encourage you to be a bit more mindful in the way these different concepts are used through the text. I also want to strongly recommend that you explicitly define in the introduction the terminology that you are going to use through the text; do not just assume the reader knows the way you personally like to refer to things. ☹️ Tweak and refine as to be precise with what you mean, and double check for consistency.

We have carefully revised the language of the manuscript to consistently use well-accepted and well-known cladistics definitions in places where ambiguity may have been present; this should also preclude the need to explicitly state these definitions, as they

are basic components of phylogeny, and general biological knowledge frequently used within the field and body of literature investigating these questions. “Stem cyanobacteria”, “stem group”, and “stem lineage” are not our invention, but refer to well-established, specific concepts in cladistics. A stem lineage refers to all ancestors of a crown group tracing back to the divergence from an extant outgroup; stem groups refer to all extinct clades along the stem lineage (for a detailed discussion of stem groups, see Budd & Mann, 2020 (DOI: 10.1126/sciadv.aaz1626)). The “total group” is inclusive of both stem and crown groups. That being said, we have improved the consistency with how we use these terms in the manuscript, and provided clarification where needed.

Assumptions

This leads me to an important aspect of the way the data is interpreted. There is an implicit assumption that oxygenic photosynthesis originated after the divergence of melainobacteria. This is by no means a fact. That the last common ancestor of cyanobacteria and melainobacteria was non-photosynthetic is not a fact, it is only a hypothesis. When we bring the evolution of the photosystems into the picture, then it is also quite possible that oxygenic photosynthesis predates what you call “stem cyanobacteria” [1]. A recent analysis of the origin of oxygen-using enzymes is also consistent with photosystem evolution [2], echoing results from nearly a decade ago [3].

I do not think it is necessary for you to discuss these aspects in your paper in any detail. However, I do think you should acknowledge and state your assumption in a direct, clear and explicit way in the introduction, discussion, or where you find it more suitable.

The reviewer is correct, our analysis and interpretation does implicitly assume that oxygenic photosynthesis evolved within stem cyanobacteria, post-dating the divergence from extant nonphotosynthetic groups. We have added language to make this assumption more explicit in the Discussion text, and in the description of Figure 4, as well as other places in the manuscript where called for:

Lines 296-310:

“These molecular clock analyses do not directly reveal how long cyanobacteria were oxygenic before their crown diversification, because species trees do not currently resolve the evolutionary transitions between intermediate physiological states such as the integration of two photosystems with chlorophyll, or the incorporation of an oxygen-evolving complex. Any groups containing these intermediate sets of characters are either extinct, or remain undiscovered. Under the additional and most parsimonious hypothesis that oxygenic photosynthesis originally arose in the cyanobacterial stem lineage following divergence from a nonphotosynthetic sister group (specific scenarios are discussed in [53][54][55]), the age of total-group Cyanobacteria serves as an older-bound on the origin of oxygenic photosynthesis. Alternatively, studies of individual photosystem genes have led to other, more complex scenarios, that hypothesize oxygenic phototrophy as more ancestral within bacterial diversity[21]. Under this proposed schema (Figure 4), these dates suggest that a diverse and ecologically robust microbial ecosystem existed on Earth long before the evolution of oxygenic phototrophy.”

Figure 4 now has the title: “Proposed narrative for the cyanobacterial context of Earth’s oxidation”.

As the reviewer notes, some previous studies [1,2] suggest that aerobic respiration predates the GOE, which would logically presuppose the existence of oxygenic

photosynthesis at this time. However, the analyses performed in [2] and [1] depend on primary and secondary molecular clock inferences that are not rigorously implemented. In particular, the analysis in [2] introduces strong taxon sampling biases, uses broad taxonomic clustering with 50% consensus rules rather than phylogenetic inferences to trace the antiquity of gene families, and relies on averaged mean secondary age estimates for dating. While critiquing this work lies outside of the scope of our manuscript, the methods we present in our study would largely remedy these issues, thereby enabling greatly improved age estimates for aerobic metabolisms by future studies. As the reviewer notes, the inferred ages of aerobic lineages and metabolisms are directly relevant to inferring the presence of oxygen in Archean ecosystems; but further evaluating the molecular clock evidence for Archean aerobic metabolisms is an entirely parallel argument to the work we present here, and is not necessary for the interpretation of these primary research results.

Other comments

Line 53, "Much of this incongruity". How much? I personally think the incongruity is mostly due to what you said in lines 48 to 50: a combination of factors. Perhaps rephrase?

We have rephrased to avoid any assumption of quantitative impact:

Lines 54-57:

"These analyses vary in the sets of fossil calibrations used to establish minimum ages for cyanobacterial groups, including some very simple forms that may or may not represent extant members of crown group cyanobacteria[23][24][25][13]."

Line 72, do you mean SI text 2? It is not clear. It would be a nice gesture for the readers if you could perhaps link the narrative of the SI text in a more fluid way to the main text. There is an abrupt change of pace when you shift from main to SI text, in such a way that I am confused as to whether you sent me to the right place: in particular, SI text 1, 2, and 5. Just add an introductory or linking opening statement. Thank you.

It would be nice too if you give the SI texts titles that reflect the content of the text.

The reviewer is correct, we have fixed this and changed it to SI text 2. We have also added descriptive headers to the SI text sections, per this suggestion. We have also edited the SI texts in some places to have more clear introductory statements where these were previously lacking.

End of introduction and start of results section. As a matter of convention, it is helpful to have a little paragraph at the end of the introduction that links to the results section, telling in a few brief statements what you did, your rationale, and/or what you found. This is particularly important given that the introduction is not followed by the methodology.

We now provide a linking paragraph at the end of the Introduction:

Lines 97-104:

"Here, we present molecular clock analyses that test the impact of different sets of fossil calibrations on species tree divergence time estimates, including the inclusion of plastid lineages and different evolutionary rate models. We also demonstrate how the use of HGTs can improve

model selection criteria and impose more stringent constraints on posterior subsampling for divergence time estimates. These analyses provide consistent support for an Archean origin of crown Cyanobacteria, and a long history of Archean biogenic oxygen production, as well as more precise age estimates for other key divergences in bacterial evolutionary history.”

The results section also opens in a way that it is not clear what is the question that you are going to address, what is the system that you are working with, or the hypothesis that you will be testing. As it is now, it is just really hard to penetrate and you will have the reader jumping all over the place trying to deduce or guess what you mean or did.

The inclusion of the linking paragraph at the end of the Introduction (see above) now addresses these questions. The beginning of the results section follows directly from this, and now clearly points to all four sets of conditions tested for their impact on cyanobacterial divergence time estimates.

Line 116, the first data you refer to in the text is SI Table 1. When I went to find that table, I found DD, BH, +older bounds, bd, no bd. At this point none of these have been introduced to the reader. I am left then trying to find what these are: it took me a while to find out that the key with the experiments is found in SI Table 3, which also use some code of white, grey, filled boxes with numbers or an x, with no explanation.

We thank the reviewer for noting that this table lacked a description; we now include this:

SI Table 3

“Different calibration/tree schema are listed in the “Key” column; “B” models include organelle sequences in the species tree (mitochondrial and plastid), while “D” models do not. Grey boxes indicate included calibrations, white boxes indicate excluded ones. Specific ages (Ga) are indicated for the endolithic fossil calibration because different fossil calibration ages were used for this group under different schema. Detailed information on each calibration is included in SI Table 2. Mito=mitochondrial group.”

I encourage the authors to reconsider how the data is presented, so that it makes for a smoother reading. To begin with, it is better if the first data that you choose to show is in the main text and not in the supplements. Perhaps you can structure things such that you first direct the reader to Table 1.

Your first paragraph of proper results, starting in line 111, is also a bit messy and hard to disentangle. Consider presenting the data first, then presenting the interpretation. For example, “under uncorrelated branch rate models the ages are X and Y. In contrast, under autocorrelated models the ages are A and B. We think these differences are caused by factors M and N, as proposed in reference [16].”

We have rewritten this first section of results per the reviewer’s suggestion, and also to introduce main text data before data presented in SI:

Lines 117-144:

“(1) Including plastid lineages and calibrations substantially impact cyanobacterial age estimates

Inclusion of red and green eukaryotic algal fossils as younger-bound calibrations had only a minor impact on crown cyanobacterial divergence times (Table 1). However, these

analyses also yielded very old ages for crown Archaeplastida, with mean ages ranging across models from 2.62 Ga to 1.97 Ga. These age ranges were substantially older than the oldest ages estimated for crown Eukarya in fossil-calibrated eukaryal molecular clock studies that included similarly old algal fossil constraints (~1.8 Ga)[37].

Even in the absence of eukaryotic fossil calibrations, the inclusion or omission of plastid sequences within the species tree substantially impacted divergence time estimates (SI Table 1) suggesting that the long branches within plastid lineages may result in an over-estimation of divergence time estimates[16]. This effect was most pronounced for uncorrelated branch rate models, with mean ages of 2.58 Ga vs. 3.07 Ga under a uniform (no_bd) prior, and 2.81 Ga vs. 3.17 Ga under a birth-death (bd) prior (Table 1). Autocorrelated models were more robust to the inclusion of plastid lineages. For example, under the CIR_nobd model, including plastid sequences increased mean age estimates by only 95 Ma (2.69 Ga vs. 2.76 Ga), supporting the consistency and reliability of autocorrelated rate models when estimating divergence times within this dataset.

This comparison suggests that placing a secondary older-bound age constraint on Archaeplastida can enable cyanobacterial divergence time studies to make best use of eukaryotic paleontological data. In fact, in the absence of cyanobacterial fossil calibrations, constraining crown Archaeplastida to be younger than 1.8 Ga had a large effect on the observed crown age of Cyanobacteria across models. Compared to analyses that used algal fossil calibrations only as younger bounds (Table 1), this additional constraint pushed mean crown Cyanobacteria ages 213-338 Ma younger under autocorrelated models (2.53 Ga - 2.76 Ga), and up to 759 Ma younger under uncorrelated models (2.44 Ga – 2.51 Ga).”

We have now also combined section (4) with the following section, to more clearly link HGT-based evolutionary model selection to the preferred set of age estimates for Cyanobacteria, as well as the increase in precision enabled by imposing HGT-based constraints on posterior sampling of trees:

Lines 215-268:

“(4) HGT-based constraints increase precision of divergence time estimates

To provide an empirical basis for the selection of evolutionary models, we used the relative ages of clades independently established by HGTs. We identified 34 strongly supported, manually curated “index” HGTs between four phyla in the species tree used for this study. These HGTs represent 24 unique donor-recipient relative age constraints within the species tree (SI Table 4, SI Figure 2). Divergence times estimated under each evolutionary model were tested for consistency with these HGT constraints, in both the presence and absence of cyanobacterial fossil calibrations. These results identified the CIR-process model combined with a uniform prior (CIR_nobd) as the most compatible in both the presence and absence of fossil calibrations. 75.79% and 65.09% of the posterior chronograms sampled under CIR_nobd fulfilled HGT constraints for uncalibrated (BH) and calibrated (BE) age estimates, respectively. Other models showed consistently lower compatibilities, between 51.83% and 58.85% for uncalibrated estimates (BH), and between 44.45% and 56.86% for calibrated estimates (BE) (SI Figure 3). The CIR_nobd model also had the most HGTs consistent with at least 20% of the posterior space sampled (21 out of 24 for uncalibrated age estimates (BH), 19 out of 24 for calibrated age estimates (BE); SI Figure 4).

Only two HGTs showed zero or near-zero (<1.2%) compatibility across all models, both with crown Chlorobiaceae as the recipient clade. Following a previous study [19], we constrained the age of crown Chlorobiaceae with a younger-bound calibration based on 1.64 Ga isorenieratane carotenoid derivatives, as these lipids synthesized by a subset of extant members of this group[48] (SI Table 2). Because this inconsistency was closely associated with one of the few

calibrated nodes within our analysis, we questioned the validity of this calibration. Removing the calibration from the divergence time estimation resulted in posterior ages with far greater compatibilities across HGTs where Chlorobiaceae is the recipient (SI Figure 5). This suggests that the calibration is phylogenetically misplaced: the ability to synthesize the observed 1.64 Ga biomarkers could have arisen earlier, either in stem Chlorobiaceae or in an entirely different clade of bacteria [49].

The preferred evolutionary model (CIR_nobd) dated crown Cyanobacteria to between 3275 and 2680 Ma, regardless of the specific set of fossil calibrations used (Table 1, Figure 3). The use of either the 1.6 Ga endolithic fossil (BJ schema) or 2.0 Eoentophysalis (BE schema) calibrations independently produced similar age range estimates for crown Cyanobacteria, with mean ages of 2960 Ma (95% CI 3275-2780 Ma) and 2977 Ma (95% CI 3219-2789 Ma), respectively (Figure 2, Table 1). When only eukaryotic primary and secondary fossil calibrations were used as calibrations, they recovered substantially younger, but still very likely pre-GOE dates for crown Cyanobacteria (95% CI 2733 – 2392 Ma). These age ranges were also consistent with those observed in previous similar molecular clock analyses that used a more controversial 1.6 Ga akinete fossil calibration[19][15]. The interpretation of these fossils as akinetes is questioned[23], but our HGT-constrained age estimates show the divergence of akinete-forming heterocystous cyanobacteria from basal heterocyst-forming types (represented in our tree by Rivularia) at 1177 -1617 Ma, not inconsistent with this interpretation (Figure 2).

Age estimates for total-group Cyanobacteria were also similar under the CIR_nobd model, with the BE schema recovering a mean age of 3468 Ma with a 95% CI age range extending to 3716 Ma, and the BJ schema recovering a mean age of 3450 Ma, with a 95% CI age range extending to 3824 Ma. Finally, imposing index HGT constraints on the subsampling of chronograms used to calculate posterior ages increased the precision of age estimates across the tree, and shifted the mean ages for most nodes substantially younger (Figure 2, Table 2). For the BE schema, the resulting mean ages of crown Cyanobacteria under the CIR_nobd model shifted 87 Ma younger to 2900 Ma (95% CI: 2958 – 2774 Ma) (SI Figure 6), and total group Cyanobacteria shifted 81 Ma younger to 3387 Ma (95% CI 3496-3278 Ma).”

Line 120-123. It is stated that uncorrelated models independently assign higher penalties across branches inferred to be evolving at faster rates. What do you mean with “independently”? Is this something that you know from how the algorithms were developed or is this your rationalization of the observed effects? Can you provide a reference where these technical aspects of rate models are presented?

One major difference between uncorrelated and autocorrelated is that the former assumes an average rate for each internal node, while in the latter, rates at the internal nodes can vary to any magnitude. Is that linked to the penalties you talk about? Perhaps you want to keep it simple and elaborate on this on a supplementary text for the more specialist audience. Consider that the broader readership of Proceedings B interested in the evolution of cyanobacteria and photosynthesis may not be aware of what autocorrelated, uncorrelated, and birth-death means.

We have simplified these statements, per the reviewer’s suggestion. As we already cite detailed reviews of these evolutionary models (see references 42-47), we have opted not to further elaborate in our supplementary material, and simply report the empirical observation of which models are most consistent with the observed HGT constraints.

You use the expression, “far more robust”, what is your measurement for robustness? How much robust exactly is “far more”?

We have removed this statement.

You conclude in the same paragraph stating that autocorrelated models are more consistent and reliable than uncorrelated. When looking at the patterns of rates of evolution from the datasets provided, I noted a substantial decrease in the rates of evolution with decreasing geological time, and a zone of large scatter after the GOE. I was not able to associate the rates to taxa, given that a file with labelled tree nodes was not provided (can you make these also available?), but I wonder if you could describe and discuss the patterns of rate change that you observe in your preferred scenarios as a supplementary text. I would appreciate that.

In the paper, we point out the consistency with respect to reporting similar ages under different taxonomic samplings (e.g., including or excluding plastid sequences) and different calibration schema. We have not included discussions of evolutionary rate changes across the tree under different models, as this was not the focus of our study. The reviewer is correct in that calculation of these rates is an important and implicit step in generating the reported divergence time estimates, and may be of particular interest to some researchers. We now provide sample treefiles with internal nodes labeled for species trees B and D (SI Dataset 5), so that readers can map the rates in the provided .dates files for each molecular clock analysis (also within SI Dataset 5).

Section “Impact of eukaryotic calibrations”. I think the data and interpretation presented in this section can be better understood in terms of effects on the rates of evolution, rather than effects on divergence times. It is known that eukaryotic sequences evolve relatively faster than those in cyanobacteria, as clearly seen in the trees you provided in Dataset S1: the “long branches within plastid lineages”. **When you add or remove taxa and calibrations as you describe, do these translate into increases or decreases on the rates of evolution?**

As correctly inferred by the reviewer below, these do, in fact, result in net increases in the rate of evolution across branches, translating into younger divergence times within Cyanobacteria. We have chosen to consistently keep the focus on divergence times in reporting results throughout our manuscript, as these are the “final product” of both rate models and calibrations (see our comment above).

For example, lines 141 to 150. In the absence of cyanobacterial calibrations and constraining photosynthetic eukaryotes to younger than 1.8 Ga, it is expected that this will translate into an acceleration of the rates of evolution for cyanobacteria, because the calculation of the rates will be dependent on the “younger than 1.8 Ga calibration”, which is allocated to the faster evolving eukaryotic sequences embedded within cyanobacteria. Therefore, it is expected that the measured ages for crown Cyanobacteria will appear younger given the calculated faster rates of amino acid substitutions.

The reviewer is correct; and this is precisely how the divergence time estimates are implicitly generated. However, as stated above, we chose to consistently focus on the resulting impact on divergence times for the sake of clarity--in this manuscript we are not specifically interested in the evolutionary rates themselves, although that is certainly its own interesting series of observations that can be extracted from our provided data. However, the impact of calibrations, like the 1.8 Ga older-bound secondary constraint, can differ, depending on how “active” such a calibration is. For example, we show that

autocorrelated models are less impacted by this older-bound constraint, presumably because they are already assigning higher rates to the plastid branches, without these being forced by the calibration. But yes, the reviewer is correct in that older-bound constraints, when included, will generally increase inferred evolutionary rates, and decrease inferred divergence times.

Consider rephrasing for clarity.

Line 137. "Far more robust". You make it sound like timing the evolution of eukaryotes with molecular clock is a done deal, or that the fossil record of eukaryotes is well understood. I am sure you recognize that molecular clocks have also generated massive uncertainties of nearly a billion years for crown eukaryotes. And that there are a bunch of eukaryote-like fossils and trace fossils that are older than 2.0 Ga. Perhaps you want to be a bit more cautious about how you make an interpretation of the utility of including plastid lineages in this type of analysis.

We have removed this statement about robustness from the manuscript. The additional fossils mentioned by the reviewer are highly controversial in their interpretations and cannot be confidently assigned as calibrations to any specific groups (see Javaux & Lepot, 2018: doi.org/10.1016/j.earscirev.2017.10.001), and, we have endeavored to only use fossil calibrations, primarily or secondarily, that are generally accepted as diagnostic. To the reviewer's broader point, we now include a mention of the potential impact of future fossil finds in the discussion.

In any case, it is not clear what the take-home message of this section is regarding timing the evolution of cyanobacteria. Perhaps you wish to restate or rephrase for additional impact.

We have revised this section to more clearly link to the implications for the timing of cyanobacteria (see the previous response above, showing our updated text in "(1) Including plastid lineages and calibrations substantially impact cyanobacterial age estimates")

Line 152, Section (2). I think enabling these new calibrations is amazing. I wonder if you have images of the microbial mats where the MAGs came from, or microscopy images of the morphological analogues of *Eoentophysalis*, or the tightly coiled cyanobacteria strains? It would be great if you could share some of these in the supplementary materials.

We thank the reviewer for their appreciation of this new data. These images have already been published in previous work, and are cited within the manuscript (Moore et al., 2020; doi:10.1130/G47394.1).

Section starting in line 224. I want to say that I am indeed very positively impressed by the HGT-enhanced clock results and the effect the approach has on narrowing the confidence intervals.

We thank the reviewer for their kind comment. We believe that this is one of the key novel contributions of this work, both for this specific application, and towards a more general goal of improving microbial molecular clock methods.

I think it would be very useful to the reader if you can provide in a separate supplementary figure, the HGT transfer events that were studied, overlaid over Fig. 2. I know it is a bit of a tedious task, but a picture is worth a thousand words!

We agree, and now provide such a figure (SI Figure 2).

Line 234, “the cumulative posterior tree space explored”. It is not immediately clear what you mean. Can you express these results in a less technical manner?

We have now changed this to “the posterior chronograms sampled” to be more clear, and more consistent with how these are now described in the updated methods section.

Paragraph starting at line 241. It is suggested that the 1.64 Ga biomarker calibration is incompatible with the HGT constraints where Chlorobiaceae is recipient. Then, you use this result to suggest that the biomarkers were not produced by Chlorobiaceae. But it is not clear why this is the case or what is the nature of this incompatibility.

The calibration is a younger bound, implying that crown Chlorobiaceae is older than 1.64 Ga, and that is exactly what I see in Fig. 2, for example. Am I wrong? So, it is not immediately obvious why the calibration is incompatible. Can you elaborate on this, perhaps on the supplements? I would appreciate being able to see in a figure how the specific transfer events are incompatible with the dates produced in the presence of the calibration for this particular case, and then made compatible once the new dates are obtained in the absence of the calibration.

The reviewer is correct. The calibration, as implemented, constrains crown Chlorobiaceae to be older than 1.64 Ga, based on the taxonomic distribution of extant Chlorobiaceae that produce these biomarkers. The chronogram presented in Figure 2 is for the evolutionary model most consistent with the relative age constraints imposed by the set of index HGTs (CIR nobd), even though this model is also incompatible with the two Chlorobiaceae-recipient HGTs identified in the text. The information the reviewer is seeking is present within SI Figure 5: Compatibility of HGT constraints and the interpretation of lipid biomarkers for Chlorobiaceae. It shows the compatibility of divergence time estimates with each index HGT, with (panel A) and without (panel B) the calibration.

We have also updated our results section to include a more clear justification for our interpretation of the misplaced 1.64 Ga isorenieratane calibration:

Lines 232-243:

“Only two HGTs showed zero or near-zero (<1.2%) compatibility across all models, both with crown Chlorobiaceae as the recipient clade. Following a previous study [19], we constrained the age of crown Chlorobiaceae with a younger-bound calibration based on 1.64 Ga isorenieratane carotenoid derivatives, as these lipids synthesized by a subset of extant members of this group[48] (SI Table 2). Because this inconsistency was closely associated with one of the few calibrated nodes within our analysis, we questioned the validity of this calibration. Removing the calibration from the divergence time estimation resulted in posterior ages with far greater compatibilities across HGTs where Chlorobiaceae is the recipient (SI Figure 5). This suggests that the calibration is phylogenetically misplaced: the ability to synthesize the observed 1.64 Ga biomarkers could have arisen earlier, either in stem Chlorobiaceae or in an entirely different clade of bacteria [49].”

I think you should also briefly mention and discuss the consistency of your best scenario with the Proterozoic fossil record of heterocystous cyanobacteria, the so-called fossil akinetes, and those presented in [4], for example. I think mentioning the heterocystous cyanobacteria fossils is important because the post-GOE “crown cyanobacteria” clocks published always produce an origin of this clade around the Phanerozoic. Lastly, I think you should also highlight the consistency of your data with the fossil record of eukaryotes themselves, since there has been a case made for crown group eukaryotes being younger than ~1.2 Ga, see [5], but see this [6].

We thank the reviewer for this suggestion, and now include it in our Results:

Lines 252-258:

“These age ranges were also consistent with those observed in previous similar molecular clock analyses that used a more controversial 1.6 Ga akinete fossil calibration [19][15]. The interpretation of these fossils as akinetes is questioned [23], but our HGT-constrained age estimates show the divergence of akinete-forming heterocystous cyanobacteria from basal heterocyst-forming types (represented in our tree by Rivularia) at 1177-1617 Ma, not inconsistent with this interpretation (Figure 2).”

As far as the consistency of our results with the eukaryotic fossil record, since we use eukaryotic algal fossils as calibrations, as well as secondary constraints based on previous studies that used additional calibrations and eukaryote sequence data, to do so would be making a circular argument. That being said, published claims of crown eukaryotes being younger than ~1.2 Ga do not take into account fossils that are generally accepted as diagnostic (e.g., ~1 Ga Bangiomorpha), and, since we use this fossil, would be a priori incompatible with the work presented here. Furthermore, the interpretation of the reported 1.6 Ga fossils reported in [6] remains controversial.

This point does, however, raise a broader issue: the age estimates for any calibrated molecular clock are limited by the diagnostic fossils we have. Future finds may increase the age of these fossil horizons, and subsequently increase the estimated ages of groups in the tree. Since these Proterozoic microfossils are so rare (or, in many cases, even singular finds), we have no way of estimating how likely these future finds may be, and extant fossils are potentially much younger than the morphotypes (and clades) they represent. However, future finds that extend the fossil horizon for these groups can only increase these age estimates, providing even stronger evidence for Archean cyanobacteria and oxygenic photosynthesis. In the case of future eukaryotic algal fossil finds that increase the age estimates for Archaeplastida, these would actually likely serve to increase consistency with the cyanobacterial fossil record, as our analyses show that chronograms using only eukaryotic fossil calibrations on plastid lineages (SI Table 1) produce ages ~200 Ma younger than those also including cyanobacterial fossil calibrations (Table 1). We now mention this issue in the Discussion section:

Lines 312-315:

“Finally, since the primary fossil calibrations used here are only younger-bounds, future discoveries of even older diagnostic fossils may provide additional evidence for crown Cyanobacteria and oxygenic photosynthesis during the Archean.”

Additional comment on HGT. There is a lot of talk in the evolution of photosynthesis about the acquisition of the process via HGT. I am quite puzzled by the fact that your HGT retrieval process did not detect or include any photosynthesis genes. There are well known transfer events of chlorophyll synthesis genes between stem Chlorobi and stem Chloroflexi... unclear direction, but I am puzzled by the fact that none of these made it to your final dataset. It has been suggested that photosynthesis was only acquired by the Chloroflexi about a billion years ago, yet you did not seem to identify any events consistent with that. It has also been suggested that cyanobacteria could have obtained photosynthesis via HGT after the divergence from melainabacteria, and so I would have expected that such events would be detectable through your retrieval methodology. Can you comment on this?

We checked back to our list of possible nested HGT candidates detected by our distance-based method, and indeed, did not find any photosystem or chlorophyll/bacteriochlorophyll synthesis protein families within that set. Our HGT detection method was designed to efficiently identify HGT events of unambiguous direction, with clear donor and recipient clades in each case. The vast majority of HGTs do not meet these “index HGT” criteria, even if they are important from the perspective of microbial evolution (e.g., photosystem genes). Our approach is agnostic to the function of the HGT genes detected; in fact, the manual curation of the final index HGT dataset was performed “blindly,” i.e., without my knowledge of the function or identity of individual genes, to prevent unconscious bias in the expert curation. Also, our initial screening of HGT candidates uses a method based on distance matrices. This method is unlikely to recover gene families containing groups separated from one another by long branches, or that have sparse distributions across the species tree (as is the case for photosystem genes). Individual HGTs detected via this method may, in themselves, contribute to our understanding of specific evolutionary narratives, but here we use HGT constraints in the abstract, as checks on evolutionary models, and constraints on age estimates rather than tracers of particular metabolisms/and physiologies.

Lines 284 to 288. You say: “Similarly constraining the older-bound estimate on the origin of oxygenic photosynthesis...” As I discussed in the Assumptions section above, here you are making some assumptions that are unproven. You are not measuring or constraining the origin of oxygenic photosynthesis. You are talking here about the node that represents the last common ancestor of melainabacteria and cyanobacteria. The data that you present do not inform on the most likely ancestral metabolic capacity of this node. Please, rephrase for precision.

As described in a previous comment above, we have updated our discussion section to be clear that older-bound age estimates for oxygenic photosynthesis are explicitly conditional on our assumption of this evolutionary narrative.

Lines 318 to 328. This is a bit confusing. In lines, 320 to 325 you say that the transitions towards the origin of oxygenic photosynthesis or the emergence of a photosystem capable of splitting water to oxygen cannot be resolved with a species trees. However, you do seem to be sure that these transitions occurred after the divergence of melainabacteria. How come?

With apologies, we do not see the source of the confusion here; multiple evolutionary intermediates must have existed as these characters were acquired, and extant taxa showing subsets of these characters are not observed today; in the species tree, we simply have crown group Cyanobacteria, which have ALL of these evolutionary innovations, and the nonphotosynthetic outgroup, which have none. Therefore, the extant species tree lacks the resolution to order the steps by which these genes and physiological processes became established in stem Cyanobacteria, before the diversification of the crown group.

Lines 338-340. Why is it likely that they represent extinct groups? Why is it not more likely that we have not completely described the diversity of cyanobacteria? I give you some examples... the discovery of *Gloeomargarita*, which might be common in hot springs but completely unknown just a few years ago. Another example, *Aurora vandensis*... then it just turned out that this has a sister lineage that lives in association with hornworts [7]! Who would have thought? Other example outside cyanobacteria, phototrophic Gemmatimonadetes that are indeed quite abundant, or Chloroflexi with type I photosystems. And these are serendipitous discoveries, not being born from projects that were dedicated to the mapping or description of a group's diversity.

We have edited the text to include the possibility that such groups survive:

Lines 300-302:

“Any groups containing these intermediate sets of characters are either extinct, or remain undiscovered.”

Lines 316-321:

*“Our study suggests that the deepest branching extant cyanobacterial lineages (*Gloeobacterales* and the A/B' lineages of thermophilic *Synechococcus*) likely diverged by 2.7 Ga. Based on the estimated ages of the deepest branching extant cyanobacterial lineages (Figure 4), the 2.7 Ga fossils and stromatolites attributed to benthic filamentous cyanobacteria[56][4] likely represent extinct or undiscovered groups that are distinct from later-diverging known groups of filamentous cyanobacteria.”*

Lines 349-351, but also SI text 5. “being early anoxygenic phototrophs of a likely extinct bacterial lineage...” How can you tell the difference between an extinct lineage of anoxygenic phototroph and an extant lineage of non-photosynthetic bacteria whose ancestors have been losing photosynthesis starting over three billion years ago? When gene loss is such a common and constant process. By what method or measurement do you distinguish between those two possibilities? How can you even tell that this is actually likely?

*The reviewer is correct that gene loss is a common process, and that many microbial clades show a diversity of energy metabolisms as a result of novel acquisitions of genes required for these metabolisms, as well as the loss of other genes and metabolic processes (e.g., *Halobacteriales* losing methanogenesis, and gaining aerobic metabolic pathways (Evans et al., 2019, <https://www.nature.com/articles/s41579-018-0136-7>). However, photosynthesis appears to be exceptional, in that no members of major extant clades of phototrophic bacteria are known to have lost it (i.e., phototrophy does not appear to be paraphyletic within major groups of bacteria, save for marine Alphaproteobacteria and Gammaproteobacteria, where it may be a relatively recent*

acquisition across many lineages independently). While possible, a scenario requiring widespread loss of photosynthetic genes in the ancestors of extant clades that are non-photosynthetic today would therefore be unexpected, and we are unaware of any phylogenomic data that support such a scenario.

How can you tell that the ancestor represented by the oldest node in Fig. 2 did not have the capacity for photosynthesis when you have phototrophic lineages on all parts of this tree, without showing us or addressing the evolution of the components of photosynthesis? I could argue that the phylogeny of type I photosystems, which features similar distances and rates between Chlorobi and cyanobacteria as you show here for your concatenated sequences, indicate that their last common ancestor had type I photosystems.

We agree that the long evolutionary distances separating different clusters of photosystem genes found in different bacterial phyla today indicate that these genes are very ancient and predate the age of extant groups of microbial phototrophs. Indeed, you could make such an argument. However, in itself, we do not see how this requires the ancestral lineages of these genes to have traversed the bacterial species tree, undergoing losses in groups lacking photosynthesis today; such a pattern could also be explained by HGT between anoxygenic phototrophic lineages throughout the Archaean and Proterozoic, with more ancient groups being extinct, creating the observed long deep branches in these gene trees. We favor this interpretation, especially in light of our response above: we do not see evidence of extant groups of phototrophs losing these genes. We now more explicitly state this assumption in the discussion text of the manuscript, as we have described in the responses above.

In addition, I believe Figure 4 is a bit misleading and misrepresents the evolution of photosynthesis. It gives the impression to the reader that evolution is a linear process. It implies that anoxygenic photosynthesis gradually evolved to become oxygenic, kind of like ferns evolving to become flowering plants, or chimpanzees evolving to become human. In reality, the early evolution of the photosystems, the complexes that enable photosynthesis, is best described as a deep dichotomy, a deep bifurcation leading to those lineages of photosystems used in anoxygenic and those used in oxygenic photosynthesis. These early stages of photosystem evolution cannot be mapped out onto a species tree of bacteria, as photosystem phylogenies suggest that these predate the main bacterial radiation. It is usually assumed [not based on any particular solid evidence, but on decades-old and weak rationales], that the ancestral states of the earliest photosystem were anoxygenic, but these views have been challenged.

We agree with the reviewer that evolution is not a linear process, insofar that we gather their meaning: that one form does not evolve into another in a linear succession of species; however, we respectfully disagree that, as depicted, the figure depicts a misleading impression of the evolutionary process. It is quite common in the depiction of evolutionary narratives to polarize the acquisition of traits with respect to a crown group (generally, this is done along the stem lineage of a cladogram). Furthermore, Figure 4 even shows this diversification, with hypothesized anoxygenic stem cyanobacteria diversifying in the Archaean, with one lineage going on to acquire oxygenic photosynthesis. We now clarify this further in the Figure 4 legend: "This hypothesized history of Cyanobacteria shows oxygenic forms arising from within a diversity of anoxygenic stem groups in the Archean."

Per the reviewer's second statement, we have addressed this issue in the responses above, and now clearly state this working hypothesis as such, and point out that other proposed narratives informed by the evolution of other genes also exist in the literature.

I would like you to compare your data with that of ref. [2]. The same node that in your tree you call "crown bacteria" Jabłońska and Tawfik named "last universal oxygen ancestor". The authors traced back to this point a major radiation of oxygen-using enzymes. They state this node is 3.1 Ga old based on the data compiled by TimeTree, but it is equivalent to your "crown bacteria", which you calculate to be about 3.6 Ga old. A similar pattern of evolution was noted by [3], in which they correlate the bacterial expansion of diversity with the availability of oxygen.

We have opted not to compare our results with this work, for the reasons described above in a previous response, namely (1) we have serious concerns about the methodological soundness of [2], and (2) a discussion of the phylogenomic evidence for the antiquity of oxygen-utilizing metabolisms, while related to this broader set of questions, is a distinct line of evidence from what we present and discuss here. Ideally, if there were more rigorous, updated molecular clock analyses of aerobic lineages and metabolisms, these would be appropriate to synthesize for a more comprehensive understanding of the dynamics of oxygen in the Archean. In fact, as we previously state, it is our hope that the HGT-based methods we describe here can, in the future, also be applied in the service of such analyses.

The point that I want to make with these reflections, is that these bits of the discussion that I highlighted, including figure 4, represent your perception of a scenario that you think likely, based on a series of assumptions that you take for granted, and integrating only selected pieces of evidence you might be more familiar with. 🙏 I encourage you to reconsider how you present the discussion:

1) I think it is important that you discuss previous molecular clock studies on bacteria and cyanobacteria, and relate your findings to that extensive literature, before you relate them to the geochemical or microfossil record. Acknowledge and consider seriously those who have come before you.

We have focused on the reporting of our primary research results, and discuss their interpretation in terms of independently testing geochemical hypotheses. This independence is an important and novel feature of our work, as we do not include any geochemical assumptions in our priors. Therefore, the key relevance is comparing these predictions to the geochemical record, which we do. To provide context for this study, we do include a representative summary of previous age estimates reported in the literature in the introduction (to which we have added an additional reference in this revised text), as well in the results section presenting our ages for SynPro groups. Aside from the necessities of the limited space available in the paper, we assert that a detailed comparison to previously published molecular clock studies is more the purview of a meta-analysis or review article.

2) Explicitly acknowledge and state your assumptions.

3) Be extremely cautious with the statements of what you perceive to be more or less likely contrasted to a substantiated body of evidence.

Methods. Starting line 444, Application of HGT-based age constraints. This methodology is not clear at all, and I am not sure it would allow others to reproduce your work. You sampled “1k and 10k chronograms”. What is the rationale for this? What is a 1k or 10k chronogram? You mean after that many cycles in phylobayes? Remember that not all software packages work the same way.

You then say that each tree is determined to be compatible or incompatible. How is this done? What is the procedure for this assessment? What does “incompatible” actually mean in terms of divergence times?

I am not sure what you mean with “tree space per HGT”. How does the attrition curve combine constraints imposed by multiple HGTs? Do you need to modify the phylobayes algorithm to be able to do this? Do you have to input a new series of calibration files incorporating the HGT information and then recalculate a clock? How do you use the HGT info to recalculate the confidence intervals, for example?

Can you please rework this section?

We have reworked the Methods section “Application of HGT-based age constraints” per the reviewers suggestions, to improve clarity and reproducibility:

Lines 418-431:

“Application of HGT-based age constraints

HGT-constrained age estimates were generated for the BE and BH calibration schemas. As provided by the “datedist” output file from PhyloBayes, chronograms sampled after burn-in each represent a likely hypothesis under the given evolutionary model. For each HGT constraint, “compatible” chronograms show the donor being older than the recipient; the percentage of compatible chronograms was then calculated for each HGT constraint (SI Figures 3 and 4). To estimate a posterior age distribution for chronograms compatible with as many HGT constraints as possible, sets of trees passing n HGT constraints were compiled, from $n=1$ to $n=24$ (SI Figure 6). Posterior age estimates were then calculated from the $n=21$ set of trees (Figures 2 and 3). This used the largest number of HGT constraints that recovered a set of compatible chronograms large enough ($n=28$) to estimate age distributions. This method is robust against the impact of a single HGT constraint or biases arising from the stepwise addition of constraints given that each set of n constraints can include any combination of HGTs.”

The rationale for the sampling of the posterior chronograms is described in the methods section “Molecular Clocks and Fossil Calibrations”. We now add the additional statement to more clearly link the sampling of posterior trees to these values:

Lines 412-415:

“For each analysis, at least the first 20% of chains were excluded as burn-in before the sampling of prior and posterior trees and calculation of age distributions in the chronogram (1k trees for uncorrelated models, and 10k trees for autocorrelated models).”

Line 405, SI Dataset 1. The trees provided labelled Model B and D both included plastid sequences. The trees provided did not appear to have branch support values.

We thank the reviewer for catching this error; we now provide the correct Model D tree, with organelle sequences removed. Our maximum-likelihood trees generated for the molecular clock analysis did not include branch support values, as tree uncertainty is not factored into these divergence time estimates. The species trees used in this paper are nearly identical to those reported in Moore et al., 2019 (which was the basis for this study), and support values are presented and discussed there:
<https://doi.org/10.3389/fmicb.2019.01612>.

Cited references

1. Oliver, T., P. Sánchez-Baracaldo, A.W. Larkum, A.W. Rutherford, and T. Cardona, Time-resolved comparative molecular evolution of oxygenic photosynthesis. *Biochim. Biophys. Acta*, 2021: 148400. DOI: <https://doi.org/10.1016/j.bbabi.2021.148400>.
2. Jabłońska, J. and D.S. Tawfik, The evolution of oxygen-utilizing enzymes suggests early biosphere oxygenation. *Nat. Ecol. Evol.*, 2021. DOI: 10.1038/s41559-020-01386-9.
3. David, L.A. and E.J. Alm, Rapid evolutionary innovation during an Archaean genetic expansion. *Nature*, 2011. 469: 93-96. DOI: 10.1038/Nature09649.
4. Pang, K., Q. Tang, L. Chen, B. Wan, C. Niu, X. Yuan, and S. Xiao, Nitrogen-fixing heterocystous cyanobacteria in the tonian period. *Curr Biol*, 2018. 28: 616-622 e1. DOI: 10.1016/j.cub.2018.01.008.
5. Porter, S.M., Insights into eukaryogenesis from the fossil record. *Interface Focus*, 2020. 10. DOI: 10.1098/rsfs.2019.0105.
6. Bengtson, S., T. Sallstedt, V. Belivanova, and M. Whitehouse, Three-dimensional preservation of cellular and subcellular structures suggests 1.6 billion-year-old crown-group red algae. *Plos Biol*, 2017. 15: e2000735. DOI: 10.1371/journal.pbio.2000735.
7. Rahmatpour, N., D.A. Hauser, J.M. Nelson, P.Y. Chen, J.C. Villarreal A., M.-Y. Ho, and F.-W. Li, Revisiting the early evolution of Cyanobacteria with a new thylakoid-less and deeply diverged isolate from a hornwort. *bioRxiv*, 2021: 2021.02.18.431691. DOI: 10.1101/2021.02.18.431691.

Referee: 2

Comments to the Author(s)

Fournier et al. combine fossil-based node age calibrations with horizontal gene transfer derived relative node age constraints to date cyanobacteria using a variety of relaxed molecular clock models.

The question is interesting and the approach applied is novel.

Major Issues:

1. How reliable are the gene trees used in the transfer inference? It seems plausible given the evolutionary time scales in question that the model used for gene tree inference (LG+G) might be inadequate for some, or indeed most, of the gene families. Given the small number of families containing the transfer of interest (<50), it would be of interest to repeat the tree inference aiming for the best fitting model for each gene family (e.g. best-fitting model in IQTree) to establish confidence in the transfers implied by the topology of the trees.

Gene trees were reconstructed using LG+G for computational expediency, as many of these gene family clusters are quite large, and model fitting each and every one of these HGT candidate gene families would take an extremely long time. Furthermore, many of the inferred HGTs only involve a very small subset of the sequences within the generated phylogenies, and best-fitting evolutionary models across the whole alignment may not reflect the models that optimally and locally recover the clades and bipartitions involved in the transfer. Therefore, a generally reliable but simple model was preferable. Having carefully considered the reviewer comments, we have added a step to the HGT vetting process to improve confidence in these results, and address the problem of substitution model fitting. We now generate additional trees for each of the selected index HGT gene families that only include the local topology containing the transfer (donor, recipient and outgroup sequences). In each case, these sequences were realigned and a phylogeny was reconstructed with the best-fitting evolutionary model in IQtree. This local reconstruction ensures that potentially highly divergent sequences elsewhere in the often-large COG subcluster trees do not have an inordinate influence on the alignment of sequences relevant to the HGT hypothesis being tested, and that the best-fitting model reflects the likelihood of substitution events directly associated with the bipartitions relevant to the HGT inference. We include these local trees in a new SI Dataset 3, and also now report the best-fitting models and bootstrap support values in SI Table 4. This extended methodology is now described in the SI text:

“SI Text 4: Detailed HGT detection methodology:

Bacterial COGs with protein to genome ratios less than 3:1 were used as the starting protein family dataset, to avoid spurious inferences from gene families with histories of extensive gene duplication. For each of the 10,453 protein families examined, patristic distances between protein sequences were transformed to network weights ($e^{-D_{ij}}$, where D_{ij} is the sum of branch lengths between homolog i and homolog j). The Louvain method for community detection[40] was applied to this network, dividing each set of homologs into closely related sub-groups for further analysis. For each phylum, distances between its members and the five closest sequences from each of the other phyla were calculated. Alphaproteobacteria were not included in HGT detection, as the large number of representatives in this dataset proved computationally intractable. Phyla represented by fewer than 5 taxa within a given sub-group were not examined. Inter-phyla distances were deemed to be potentially indicative of HGT events if the median distance to the closest phylum was significantly lower than that of the second-closest phylum ($p=0.01$), resulting in 3,949 sub-groups containing potential HGT candidates. For each sub-group identified as potentially containing an inter-phylum HGT, sequences were re-aligned using MAFFT[41] with automatic parameter selection. Trees were subsequently reconstructed using IQTree[42], under the LG+G model, and supports provided by rapid bootstrapping ($n=1000$) and an approximate likelihood ratio test ($n=1000$). Reconstructed sub-group trees were automatically rooted using Minimal Ancestor Deviation (MAD)[43] (Tria et al., 2017). 249 gene families with nested (paraphyletic) pairs of phyla indicative of index HGTs were then automatically detected using in-house scripts based on the ETE3 python library[44]. Manual vetting of these index HGT candidates was then performed, identifying 38 index HGT candidates. These candidate HGTs were further evaluated using taxon samplings that only included the local topology containing the candidate HGT (donor, recipient, and outgroup sequences), which further restricted the set of inferred index HGTs to 34 events. This local reconstruction ensures that potentially divergent sequences elsewhere in the often-large COG subcluster trees do not have an inordinate influence on the alignment of sequences relevant to the HGT hypothesis being tested, and that the best-fitting model reflects the likelihood of substitutions directly associated with the bipartitions relevant to the HGT inference. For these taxa, sequences were realigned

and trees reconstructed in IQtree using the best-fitting evolutionary model. Supports for inferred HGTs were estimated based on the “degenerate” bootstrap support for the nestedness of donor-recipient pairs, as multiple bipartitions representing specific donor-recipient hypotheses may each be consistent with the relative temporal ordering of the donor and recipient nodes. These values were extracted from bipartition tables (SI Dataset 6) and reported in SI Table 4. Specific narratives for each HGT inference are included in SI Text 5.”

3. Are gene tree branch supports (e.g. bootstrap values) on gene phylogenies taken into account? It might be that groups that appear in an unexpected position are not supported.

Branch supports are taken into account; these are now reported in SI Table 4. In particular, degenerate branch supports are reported, as described in our response to point (3) below, and also in our extended methodology (SI Text 4: Detailed HGT detection methodology).

3. Are the same transfers (index HGTs) recovered by gene tree - species tree reconciliation methods, in particular ones, such as GeneRax (Benoit 2020 <https://doi.org/10.1093/molbev/msaa141>) that are able to take into account gene tree uncertainty?

We initially attempted to detect time-calibrating HGTs using reconciliation-based methods, namely, Ranger DTL (<https://doi.org/10.1093/bioinformatics/bty314>). Extensive tests and manual examination of the predicted HGTs using this approach did not lead to consistently observed “nested” donor-recipient clades in gene trees identified as containing transfers between specific groups with high probability. We also observed many cases of clearly false inferences arising from misrootings or long branches with an absence of species-tree outgroups. We believe that this is primarily due to (1) identification of transfers in the context of several inferred “losses” that often result in trees without apparent nested donor-recipient clades; and (2) mis-rooting of trees via the assumption that the true root should minimize the number of reticulation events, especially in the absence of a well-supported outgroup. We collaborated with the authors of Ranger DTL to address the latter issue, which resulted in work that justified our concerns: <https://doi.org/10.1371/journal.pone.0232950>. This prompted us to develop a distance-based approach that does not depend upon phylogenetic reconstruction in the initial selection of likely HGT candidates--rather, it simply identifies sequences from different phyla that have much smaller distances between them than would be expected if they had evolved along a species tree--that is, sequences for which representatives from one phyla have shorter distances to a group from another phyla, than that phyla does to itself. After further automated selection of pairs of phyla with “nested” representation within each tree, the relatively small number (n=249) of candidate gene families that satisfied this criteria were then subjected to phylogenetic reconstruction and manual curation. Gene tree uncertainty was not a major factor in these inferences, since our approach favored stringency; poorly resolved gene trees where a nested HGT donor-recipient clade relationship was not readily apparent were omitted from consideration. This is aided by the “degeneracy” of our approach. Often, the exact placement of a recipient clade within a donor group has low support, but a deeper bipartition that firmly established a nested placement within a broader donor group has

very high support; these cases were identified during manual curation, and the oldest well-supported nested donor clade was selected as the constrained “older” node. This is appropriate because our goal is not to identify the most likely reticulating branch linking donor and recipient groups; rather, it is to find the clades that are most reliably temporally constrained by the observed donor-recipient transfers. Additionally, bootstrap supports for placement of an HGT recipient group may be low, but aggregate support for equivalent “sister” placements may be high: If an HGT recipient group R is placed within Clade A+B, it does not matter if R groups with A or B; either way, clade R will be constrained to be younger than clade A+B. **We now include a more detailed description of this HGT inference methodology in SI Text 4, and now include reports of degenerate support values for index HGT inference in SI Table 4, as well as the bipartition tables used to calculate these values in SI Dataset 6.** We agree with the reviewer that, in theory, a reconciliation-based method would efficiently detect index transfers. However, we assert that our selection criteria is sufficiently stringent, that secondary detection by a reconciliation method would not increase our confidence in these transfers (which we would still subject to the same manual curation vetting). It is not our intent to detect ALL inter-phylum transfers that explain the reticulate history of these genomes; rather, we wished to efficiently find the most unambiguous cases of HGT events that established the relative ages of groups from different phyla on the species tree. We believe that this approach is relatively comprehensive, as we recovered the only two other dating-suitable HGTs between these phyla that we had previously found via ad hoc methods (SahH, <https://doi.org/10.1111/qbi.12273>; and SOD, unpublished), as well as two pairs of transfers that represent HGTs of complete metabolic pathways (ThrB/ThrC), and functional subunits of larger enzyme complexes (SdhC/SdhD).

4. Removing the two transfers for Chlorobiaceae improves the general agreement with the HGTs but does this really mean that source of error is coming from the calibration in Chlorobiaceae? Could there be another possible explanation? How can we determine if the calibration or the transfers that imply the conflicting relative age constraints are more likely to be correct?

The reviewer raises an interesting and important point, and is correct that the question is one of varying degrees of belief in different sources of evidence. First, we have no reason, based on our HGT methodology, to doubt the veracity of these HGTs any more or less than the others in our analysis. The two HGT constraints implicated (1CG and 7BG) include two different donor phyla, providing independent relative donor-age constraints, further supporting that the issue is with the age of the recipient group. This recipient group (Chlorobiaceae), is also one of the few clades directly constrained by the calibrations used in this study, suggesting that the issue lies with the recipient calibration. Second, this calibration inference differs from the others in our analysis, as it depends not upon morphological fossil evidence, but preserved lipid biomarkers, and the observed metabolic capability to produce these biomarkers in extant taxa. Therefore, the application of the calibration assumes the parsimonious explanation that this trait arose within crown Chlorobiaceae. An alternative explanation would be that production of these lipids arose earlier in the history of Chlorobiaceae, within the stem lineage, but was subsequently lost by some members of the crown group. In such a case, imposing the 1.6 Ga younger-bound constraint within crown Chlorobiaceae would potentially produce ages that are too old to be compatible with this group being the recipient of some HGT, even if these are valid events. This is analogous to a common problem

within fossil calibration, wherein an apparent shared derived character (synapomorphy), assumed to have appeared within a crown group, is actually ancestral to this group, and lost or ambiguous in some deeply branching members. This can result in fossils being misplaced as shallower within the tree, resulting in an overestimation of crown group ages (for an example of how these interpretations challenge the phylogenetic assignment of fossils, see Cunningham et al., 2017. The Weng'an Biota (Doushantuo Formation): an Ediacaran window on soft-bodied and multicellular organisms, J Geol Soc London. 174(5):793–802, <https://doi.org/10.1144/jgs2016-142>). We have updated our discussion of the Chlorobiaceae calibration to more clearly communicate these issues, and justify our interpretation:

Lines 232-243:

“Only two HGTs showed zero or near-zero (<1.2%) compatibility across all models, both with crown Chlorobiaceae as the recipient clade. Following a previous study [19], we constrained the age of crown Chlorobiaceae with a younger-bound calibration based on 1.64 Ga isorenieratane carotenoid derivatives, as these lipids synthesized by a subset of extant members of this group[48] (SI Table 2). Because this inconsistency was closely associated with one of the few calibrated nodes within our analysis, we questioned the validity of this calibration. Removing the calibration from the divergence time estimation resulted in posterior ages with far greater compatibilities across HGTs where Chlorobiaceae is the recipient (SI Figure 5). This suggests that the calibration is phylogenetically misplaced: the ability to synthesize the observed 1.64 Ga biomarkers could have arisen earlier, either in stem Chlorobiaceae or in an entirely different clade of bacteria [49].”

5. The authors find that evolutionary model choice is a major determinant of divergence time estimates, but only investigate different models for the evolutionary rates (UGAM, LN and CIR) and for the divergence time prior (birth-death vs uniform), but do not investigate what effect the choice of substitution model may have. In particular, does using CAT-GTR in the RMC dating step have a substantial effect compared to the C20 mode used? (The number of mixture components in site-heterogenous models can have a strong effect on branch lengths, cf. FIG. 3. in Schrepf et al. 2020 <https://doi.org/10.1093/molbev/msaa145>)

The reviewer is correct, we do not explicitly test the effect of substitution model (including CAT-GTR vs. catfix models such as C20) and instead focus on the impact of evolutionary models given a tree. This, in itself, is an area of major discussion in the field. We thank the reviewer for bringing this recent paper to our attention. In examining Fig. 3 in Schrepf et al. 2020, and the associated results in the text of this paper, we do not consider the effect demonstrated there to be broadly applicable to our kind of presented investigation. First, only the overall sum of branch lengths is measured, and so from these results one cannot determine if this is due to a general increase in branch lengths across the tree topology, or from the contribution of specific lineages sensitive to the increase in CXX categories. In a relaxed molecular clock model, branch length increases scaled uniformly across a tree will not impact divergence time estimates, and will only increase the rates fit to branches. The differential impact of branch lengths across a tree is more important and can result in local relative rate changes that could impact divergence times. Second, this example is based on test datasets that were explicitly selected because they demonstrate clear cases of long branch attraction between small groups of taxa, that these models aim to mitigate. Therefore, observed

differences in branch length are expected to be maximized. Even so, the observed total branch lengths in the test datasets vary, at most, by about 10% in each case with respect to increasing the category counts in CXX models. We would expect that, over a much larger tree sampling lineages across the bacterial domain, the potential impact of substitution models on divergence times would be far less than what is observed in Schrempf et al.

Previous studies have investigated, in a broader sense, the impact of substitution models on divergence time estimates. For example, see Tao et al., 2020 (<https://doi.org/10.1093/molbev/msaa049>) Figures 1 and 3, which show that even a severe mis-estimation of branch lengths under a jukes-cantor model vs. a GTR model showed no significant impact on resulting divergence time estimates across multiple phylogenetic datasets, including metazoans, plants, and prokaryotes. Similar results have also been observed for mammalian datasets: Du et al., 2019 (<https://doi.org/10.1186/s12862-019-1534-9>, Figure 4).

Minor Points:

Abstract: The authors state both the precision and accuracy of dates are improved by incorporating relative age constraints. A reduction in uncertainty is expected, but how do we (or indeed can we) know that accuracy is improved? Do the authors demonstrate an increase in accuracy?

The reviewer is correct; we show consistency with HGTs, which is not the same as accuracy (as defined by predicting known ages). We have corrected this statement and no longer claim that our results show an increase in accuracy.

lines 92-94 “species tree reconciliations” gene trees are reconciled with species trees, I assume the authors mean “gene tree - species tree reconciliation methods”?

We have changed the text to use this more precise description.